# Coarsening dynamics can explain meiotic crossover patterning in both the presence and absence of the synaptonemal complex

John A Fozard[1], Chris Morgan[2]*, Martin Howard[1]*

[1]Computational and Systems Biology, John Innes Centre, Norwich Research Park, Norwich, United Kingdom; [2]Cell and Developmental Biology, John Innes Centre, Norwich Research Park, Norwich, United Kingdom

**Abstract** The shuffling of genetic material facilitated by meiotic crossovers is a critical driver of genetic variation. Therefore, the number and positions of crossover events must be carefully controlled. In *Arabidopsis,* an obligate crossover and repression of nearby crossovers on each chromosome pair are abolished in mutants that lack the synaptonemal complex (SC), a conserved protein scaffold. We use mathematical modelling and quantitative super-resolution microscopy to explore and mechanistically explain meiotic crossover pattering in *Arabidopsis* lines with full, incomplete, or abolished synapsis. For *zyp1* mutants, which lack an SC, we develop a coarsening model in which crossover precursors globally compete for a limited pool of the pro-crossover factor HEI10, with dynamic HEI10 exchange mediated through the nucleoplasm. We demonstrate that this model is capable of quantitatively reproducing and predicting *zyp1* experimental crossover patterning and HEI10 foci intensity data. Additionally, we find that a model combining both SC- and nucleoplasm-mediated coarsening can explain crossover patterning in wild-type *Arabidopsis* and in *pch2* mutants, which display partial synapsis. Together, our results reveal that regulation of crossover patterning in wild-type *Arabidopsis* and SC-defective mutants likely acts through the same underlying coarsening mechanism, differing only in the spatial compartments through which the pro-crossover factor diffuses.

*For correspondence:
chris.morgan@jic.ac.uk (CM);
martin.howard@jic.ac.uk (MH)

Competing interest: The authors declare that no competing interests exist.

## Editor's evaluation

This important paper discloses a new control mechanism of meiotic crossing over, which is essential for the segregation of homologous chromosomes. With mathematical modeling and super-resolution imaging, the work provides convincing experimental data to support a model of "nucleoplasmic coarsening" between recombination intermediates and nucleoplasm for the control of crossover distribution in the context of a meiotic chromosome structure. The work will be of interest to researchers who work on meiosis as well as the regulation of chromosomal biology in general.

## Introduction

During meiotic prophase I, pairs of threadlike homologous chromosomes are tethered together through the formation of a protein scaffold called the synaptonemal complex (SC) (*Page and Hawley, 2004*). Prior to SC formation, programmed double-strand breaks (DSBs) form along the length of the chromosomal DNA (*Gray and Cohen, 2016*; *Hunter, 2015*). DSBs are then repaired via the formation

of recombination intermediate (RI) DNA joint-molecules as either non-crossovers (NCOs) or crossovers (COs) (*Gray and Cohen, 2016*; *Hunter, 2015*). In most species, the number of RIs repaired as NCOs vastly outnumbers those repaired as COs (*Gray and Cohen, 2016*; *Hunter, 2015*). Importantly, nearby pairs of COs on the same chromosome are less likely to occur than more distant pairs, but each pair of chromosomes always receives at least one CO. These phenomena are known as 'crossover interference' and 'crossover assurance', respectively (*Jones and Franklin, 2006*; *Otto and Payseur, 2019*; *Sturtevant, 1915*; *von Diezmann and Rog, 2021*; *Zickler and Kleckner, 2016*).

Recently, we introduced a mechanistic coarsening model that quantitatively explains CO interference, CO assurance, and other CO patterning features in wild-type *Arabidopsis thaliana* (*Morgan et al., 2021*). In our SC-mediated coarsening model, the pro-CO protein HEI10 (*Chelysheva et al., 2012*; *Ziolkowski et al., 2017*) can diffuse along synapsed homolog bivalents and can reversibly bind into immobile clusters at RI sites. Conservation of the total amount of HEI10 on individual bivalents, along with an unbinding escape rate from RIs that decreases as more HEI10 locally accumulates, causes the system to coarsen. By the end of the pachytene substage of prophase I, significant amounts of HEI10 are retained at only a small number of predominantly distantly spaced RIs, each with high levels of HEI10, which are then designated to become COs (*Morgan et al., 2021*). A similar model was subsequently proposed to underlie CO patterning in *Caenorhabditis elegans* (*Zhang et al., 2021*), suggesting that this molecular coarsening paradigm represents a conserved feature of meiotic CO control.

The coarsening model fundamentally relies on the retention of HEI10 molecules on synapsed bivalent chromosomes, and redistribution of HEI10 molecules by diffusion along the bivalents. Work in *C. elegans* points to the SC acting as a conduit, or channel, through which HEI10 molecules can diffuse, promoting coarsening along individual bivalents. Consistently, in *C. elegans*, the SC is known to have liquid-crystal-like properties that can spatially compartmentalise pro-CO proteins (*Rog et al., 2017*; *Zhang et al., 2018*). Furthermore, innovative live-cell imaging experiments have recently directly demonstrated that ZHP-3, a HEI10 ortholog, can diffuse along the SC (*Stauffer et al., 2019*; *Zhang et al., 2021*). In *Arabidopsis*, it was also recently shown that CO interference and assurance are abolished in plants lacking the SC transverse filament protein, ZYP1 (*Capilla-Pérez et al., 2021*; *France et al., 2021*). These observations suggest that the SC plays a direct role in mediating crossover patterning dynamics.

In this work, we therefore develop a new coarsening model for crossover interference in the absence of an SC and use it to explore the regulation of crossover number and position in *Arabidopsis zyp1* mutants with abolished synapsis. In this new model, due to the lack of an SC, HEI10 molecules are no longer restricted to diffuse along individual bivalents. Instead, HEI10 is assumed to diffuse through a communal nucleoplasm, generating competition for HEI10 between RIs located on different bivalents. Similarly, in *C. elegans*, it has previously been suggested that the SC functions to prevent exchange of recombination proteins between RIs via the nucleoplasm (*Rog et al., 2017*; *Zhang et al., 2021*; *Zhang et al., 2018*). Other studies have also recently proposed that nucleoplasm-mediated coarsening can explain CO patterning in *Arabidopsis* mutants with abolished synapsis, although the dynamics of this process have not been explicitly modelled (*Durand et al., 2022*; *Lloyd, 2023*). Using our model, we provide a mechanistic explanation for why *zyp1* mutants without synapsis lose both CO interference and assurance. Intriguingly, we also reveal and explain an additional layer of crossover control, where *zyp1* mutants lose positional effects that are determined by CO interference along individual chromosome pairs, but still exhibit constraints on total crossover number per-cell. Importantly, we also find that a model combining nucleoplasm-mediated and SC-mediated coarsening is still fully capable of explaining CO patterning in wild-type *Arabidopsis*, as well as in *pch2* mutants, which display incomplete synapsis (*Lambing et al., 2015*). Overall, our results demonstrate a critical role for the SC in controlling and constraining the dynamic coarsening of HEI10 molecules.

## Results

### A nucleoplasm-mediated coarsening model for crossover patterning without the SC

The SC likely acts as a conduit for HEI10 diffusion along synapsed pachytene bivalents (*Morgan et al., 2021*; *Rog et al., 2017*; *Zhang et al., 2021*; *Zhang et al., 2018*). Without an SC, we postulated

that HEI10 diffusion could instead occur through the nucleoplasm, with little or no movement along chromosomes, as has also previously been suggested from work in *C. elegans* (*Zhang et al., 2021*). We therefore investigated this hypothesis by developing a new coarsening model for crossover interference without an SC. This 'nucleoplasmic coarsening' model incorporates the processes shown in *Figure 1A and B* on the five pairs of homologous chromosomes in each cell, whose lengths are drawn from the experimentally measured length distributions, and with randomly positioned RIs ('Materials and methods').

In this model, HEI10 is able to move from the nucleoplasmic pool to the HEI10 clusters at each of the RIs. It can also escape from the RI clusters back into the nucleoplasmic pool, at a rate which depends on the amount of HEI10 within that RI cluster. Similar to previous work (*Morgan et al., 2021*), this rate is chosen to have the form shown in *Figure 1C and D*, decreasing as the amount of HEI10 within the RI compartment increases. Again, as in our earlier model (*Morgan et al., 2021*), if the total amount of HEI10 is sufficiently high then a uniform steady state becomes unstable and the system progressively coarsens, with RIs with more HEI10 growing at the expense of those with less. Eventually, the majority of the HEI10 will accumulate into a single focus per cell, but the limited duration of pachytene means that this process does not complete, and instead there are a number of RIs with significant levels of HEI10 at the end of the simulation. Full details of the simulated model can be found in *Figure 1A–G* and 'Materials and methods', including minor differences in how the model was simulated (in the initial distribution of HEI10, dynamics at small HEI10 foci and criteria for HEI10 focus/CO calling) compared to our earlier work (*Morgan et al., 2021*).

## Nucleoplasmic coarsening model explains CO number distribution in *zyp1* mutants

To determine whether the nucleoplasmic coarsening model was capable of explaining CO patterning in *Arabidopsis* mutants without synapsis, we first manually adjusted model parameter values to fit simulation outputs to existing experimental data on CO frequency from *Arabidopsis zyp1a zyp1b* null mutants, which lack an SC. These data include total CO number and number of homologs without a CO (*Figure 2A and B*).

By counting the number of MLH1 foci in late-prophase I cells in seven different Col-0 *zyp1* null mutant lines, it was previously found that there was an ~50% increase in the predominant class I COs in these mutants compared with wild-type (data from *Capilla-Pérez et al., 2021*, shown in *Figure 2A*). Also, by analysing DAPI-stained metaphase I cells, it was found that ~11% of metaphase I cells contained a pair of univalent chromosomes, which form when homologs fail to form a single CO, indicating an absence of CO assurance (data from *Capilla-Pérez et al., 2021*, shown in *Figure 2B*). Importantly, we found that the nucleoplasmic coarsening model was capable of recapitulating the increase in CO number and the increased univalent frequency observed in *zyp1* mutants (*Figure 2A and B*). The number of COs per cell was slightly less dispersed, with a lower sample variance, within our model than in the experimental data. This reduction is likely because the experimental data is pooled from multiple different mutant lines, each of which has a slightly different mean number of COs per cell, thereby generating a broader distribution (*Capilla-Pérez et al., 2021*). The expected frequency of univalents is also slightly higher in our simulation outputs than in the experimental data, but this is again expected as chiasma number in metaphase I cells (that were used for the experimental analysis) are influenced by both class I and class II CO numbers, whereas the nucleoplasmic coarsening model only simulates CO patterning via the (dominant) class I pathway. Assuming approximately two additional class II COs per cell (*Mercier et al., 2005*) are distributed randomly among the five chromosome pairs, in a simple estimation we would expect the univalent frequency in our model output to be reduced by a factor of $(0.8)^2 = 0.64$ (from 19% of cells to 12% of cells containing one or more univalent). However, univalents are more likely to be associated with the shorter chromosomes as they have fewer RIs. This was recently experimentally demonstrated, with chromosome 4 (the shortest *Arabidopsis* chromosome) exhibiting the highest frequency of aneuploidy in *zyp1* mutants (*Durand et al., 2022*). This effect slightly reduces how much the addition of class II crossovers decreases the frequency of cells with one or more univalent. In our simulations, we thus find a value of ~13%, similar to but slightly higher than previous observations (~11%) (*Capilla-Pérez et al., 2021*).

Using the nucleoplasmic coarsening model, we were also able to simulate the distribution of crossovers on each of the five *Arabidopsis* chromosomes (*Figure 2C*). Consistent with previous genetic

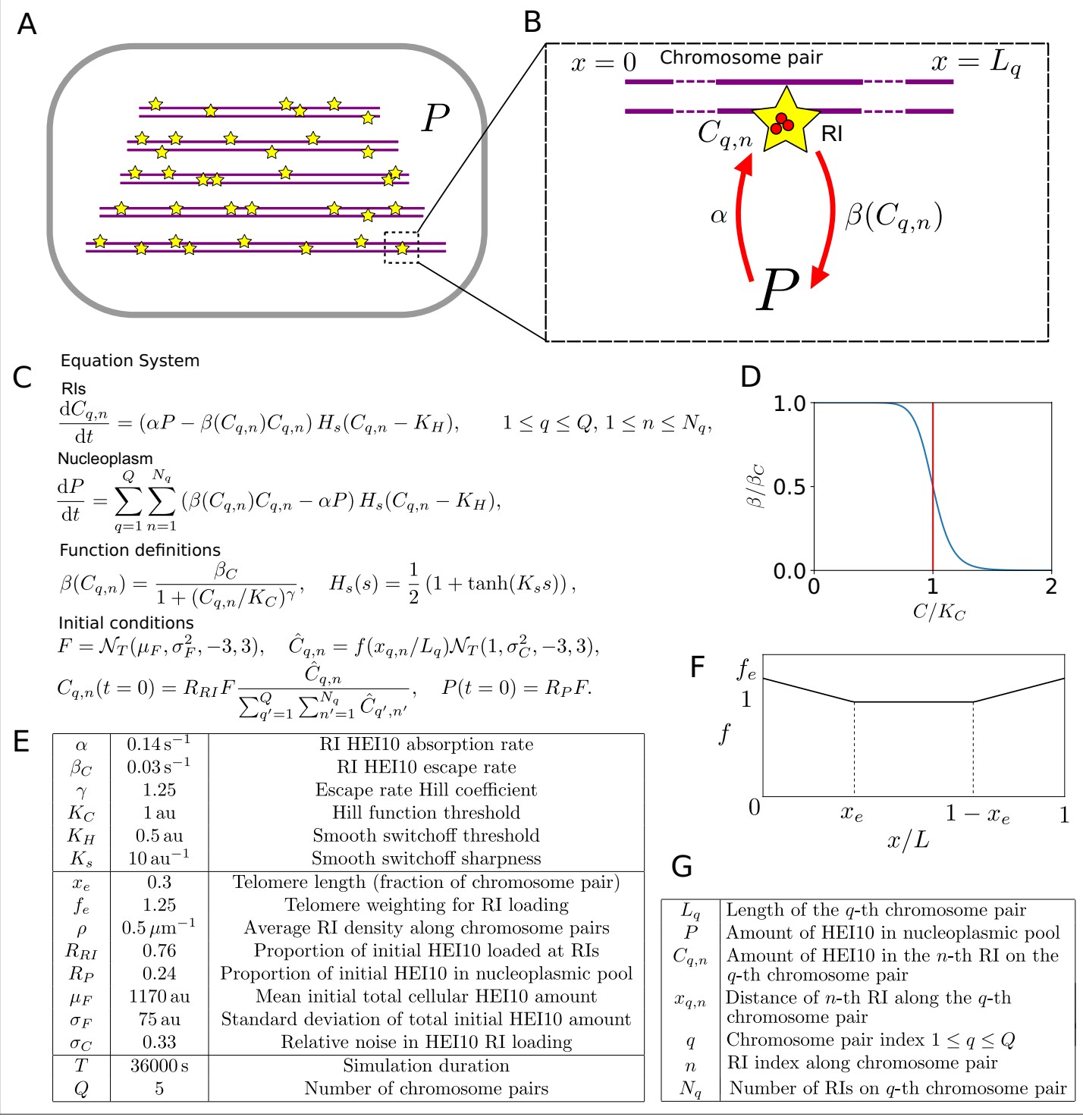

**Figure 1.** Mathematical model for HEI10 dynamics in a *zyp1* synaptonemal complex (SC) mutant. (**A**) Each cell contains Q = 5 chromosome pairs (purple line pairs). Recombination intermediates (RIs) (yellow stars) are placed randomly along the chromosome pairs immersed in the nucleoplasm (P). (**B**) HEI10 (red) is able to move from the nucleoplasmic pool into the RI compartments (rate α) and escape (rate β) from the RI compartments back into the pool. $C_{q,n}$ is the HEI10 molecule number at the nth RI on the qth chromosome pair, which has length $L_q$. (**C**) Differential equations governing nucleoplasmic pool (P) and RI HEI10 (C) amounts, together with functional form of escape rate function $\beta(C)$, sigmoidal smoothing function $H_s(s)$, and initial conditions for RI and nucleoplasmic HEI10 amounts. This sigmoidal smoothing function effectively switches off nucleoplasmic recycling for RIs with insufficient HEI10, without introducing a discontinuity in the system of equations, which would complicate numerical simulation. $\mathcal{N}_T$ is a Normal

*Figure 1 continued on next page*

*Figure 1 continued*

distribution, truncated at 3 standard deviations away from its mean. (**D**) Graph showing RI escape rate function $\beta\left(C\right)$. (**E**) Default simulation parameter values. (**F**) Form of end-bias function $f$. (**G**) Description of terms appearing in model equations.

The online version of this article includes the following source data and figure supplement(s) for figure 1:

**Source data 1.** Default simulation parameter values for nucleoplasmic coarsening model.

**Figure supplement 1.** Form of end-bias function $f$ from nucleoplasmic coarsening model (*Figure 1F*) and from the synaptonemal complex (SC)-mediated coarsening model (*Morgan et al., 2021*).

analyses, we found that the average number of crossovers per chromosome pair is positively correlated with their physical size (*Durand et al., 2022*). We also found that the distribution of the number of crossovers on each chromosome is close to a Poisson distribution, with nearly equal mean and variance. Processes that occur at a constant probability per unit space or time, when integrated over a fixed interval, have a Poisson distribution (*Haldane, 1931*). Hence, as expected, CO interference along each chromosome is not present in this model. These findings are consistent with a previous genetic analysis, which found that the distribution of CO numbers on individual chromosomes did not deviate from a Poisson distribution (*Capilla-Pérez et al., 2021*).

Paradoxically, while the distribution of CO numbers on individual chromosomes in our simulations closely followed a Poisson distribution, the total number of COs per cell did not. Applying a test for underdispersion (*Cameron and Trivedi, 1990*), the distribution of simulated CO numbers was found to be significantly underdispersed relative to a Poisson distribution ($z = -276.85$, $p<0.001$, see also *Figure 2A*). Compellingly, we found that a sub-Poissonian distribution of total CO numbers per cell was also found in the experimental data from *Capilla-Pérez et al., 2021*; *Figure 2A*; test for underdispersion $z = -6.5757$, $p<0.001$, although this experimental underdispersion was not previously highlighted. In the model, the total number of crossovers in each cell is regulated through global competition for a limited pool of HEI10. This, combined with regulation of cell-to-cell total HEI10 amounts ('Materials and methods'), leads to underdispersion despite the fact that there is little or no control on the number of crossovers on each chromosome pair. This underdispersion persists in the model even with altered initial conditions (eliminating non-uniform initial HEI10 loading, 'Materials and methods', *Figure 2—figure supplement 1*).

We also found that the coarsening dynamics within the nucleoplasmic coarsening model were capable of producing the experimentally observed final CO numbers within the limited ~10 hr duration of the pachytene substage of prophase I in *Arabidopsis* (*Prusicki et al., 2019*; *Figure 2D*). Overall, we find that the nucleoplasmic coarsening model fits well with prior data on CO numbers in SC mutants, giving strong support to the hypothesis of nucleoplasmic HEI10 exchange.

## Nucleoplasmic coarsening model explains CO number distribution in *zyp1* HEI10 over-expressing mutants

To further test the ability of the nucleoplasmic coarsening model to explain CO patterning in mutants without synapsis, we also sought to fit the model to recent experimental data examining the combined effect of *zyp1* loss of function and HEI10 over-expression (*Durand et al., 2022*). Again, by counting the number of MLH1 foci in late-prophase cells, it was found that combining the *zyp1* mutation with HEI10 over-expression leads to an average of 45.0 ($\sigma^2 = 64$) class I COs per cell in male Col-0 meiocytes (*Durand et al., 2022*; *Figure 2—figure supplement 2A*). This represents an increase in class I COs to approximately 3.5 times the number in wild-type. Unlike in the sole *zyp1* mutant, there was no evidence of a sub-Poissonian distribution of COs per cell within the *zyp1*+HEI10 over-expression lines (test for underdispersion; $z = 1.0843$, $p=0.86$).

Increasing the total HEI10 within our nucleoplasmic coarsening model to 3.5 times its original amount (applied to both the mean and standard-deviation of the distribution of total cellular HEI10) gave approximately the same increase in CO number (to mean CO number 47.4, shown in *Figure 2—figure supplement 2B*), but did not generate the observed increase in variance ($\sigma^2 = 14$), which was significantly smaller than the experimental observations (Levene's test for equality of variances $F(1,10034) = 68.602$, $p<0.001$). To obtain increased variance in CO number, we increased the standard deviation in the total cellular HEI10 levels, increasing by approximately an order of magnitude ('Materials and methods'). With this additional noise, which we attribute to presence of the HEI10 transgene,

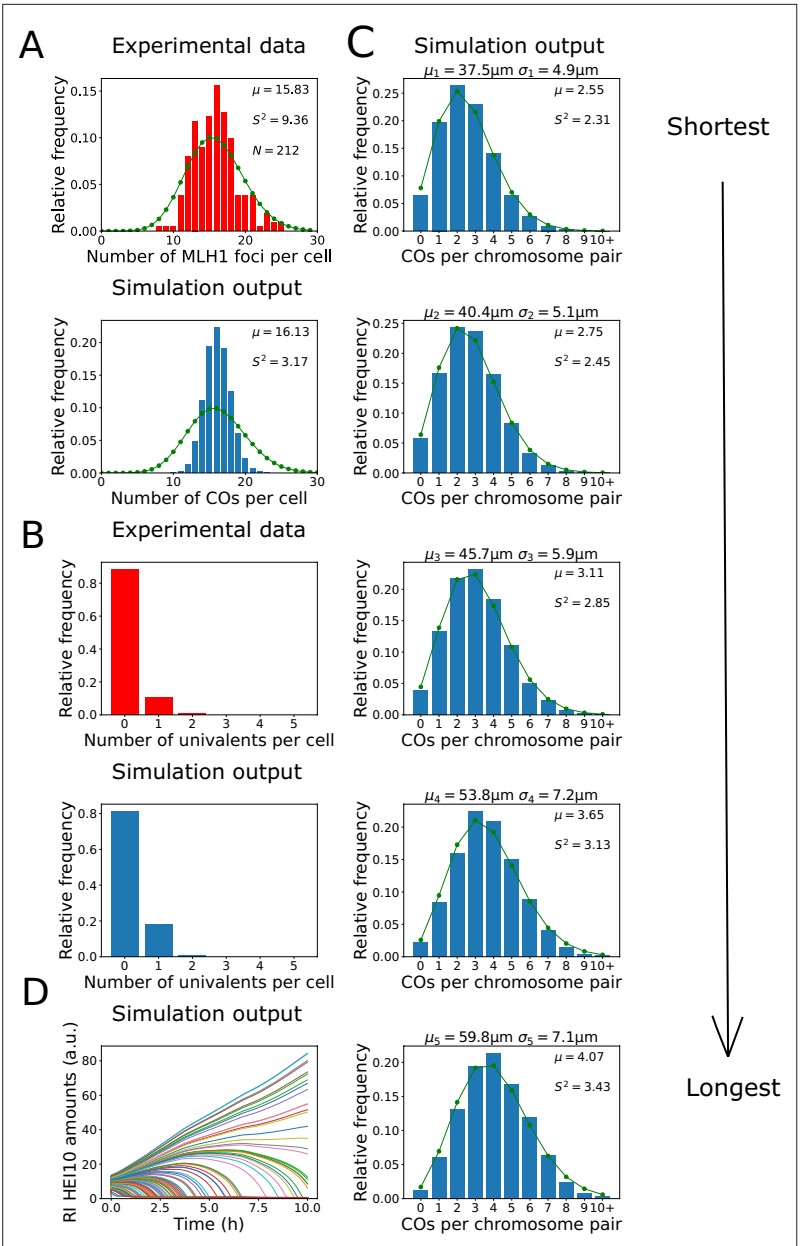

**Figure 2.** Analysis of crossover (CO) number in *zyp1* mutant. (**A, B**) Experimental data from *zyp1* null mutant plants (top) and nucleoplasmic coarsening model simulations (bottom). Results from simulating the model for 10,000 cells are shown. (**A**) Distribution of total CO number per cell. Experimental data is from pooled MLH1 foci counts from *Capilla-Pérez et al., 2021*. Sample mean ($\mu$), estimated variance ($S^2$), and sample size ($N$) inset. Green dots (joined by a line) show a Poisson distribution with the same mean. (**B**) Distribution of univalent number per cell. Experimental data is from univalent counts in metaphase I chromosomes from *Capilla-Pérez et al., 2021*. Again data from 10,000 simulated cells. (**C**) Simulation outputs showing the distribution of total CO numbers on the five individual chromosomes pairs. Green dots and lines again show Poisson distributions with same means as simulated data, with sample mean ($\mu$) and estimated variance ($S^2$) inset. Chromosome mean lengths ($\mu_i$) and standard deviations ($\sigma_i$) shown above each histogram. Simulation output from 10,000 cells. (**D**) Traces of HEI10 focus intensity against time, for all recombination intermediates (RIs) within a single simulated cell. Coloured lines indicate HEI10 amounts at each RI, in arbitrary units (a.u.). All simulation parameters are listed in *Figure 1E*.

The online version of this article includes the following figure supplement(s) for figure 2:

**Figure supplement 1.** *zyp1* total crossover (CO) number per cell simulation without nonuniform initial loading.

**Figure supplement 2.** *zyp1*+HEI10 over-expression total crossover (CO) number per cell.

we were able to recapitulate this experimentally observed increase in CO number and variance, with simulation outputs giving an average of 46.9 COs per cell ($\sigma^2 = 63$, shown in *Figure 2—figure supplement 2C*), this variance not being significantly different (F(1,10034) = 0.0982, p=0.754).

## Super-resolution imaging of HEI10 in SC-defective *zyp1* mutant cells

As described above, previous studies have used HEI10 and MLH1 focus counts per cell, alongside genetic analyses, to examine the effects of SC absence on CO frequency in *Arabidopsis*. However, the material positioning of recombination sites (in units of microns) along prophase I chromosomes has not yet been cytologically investigated in *zyp1* mutants. This is particularly important because, as argued in *Zickler and Kleckner, 2016*, the optimal metric for measuring CO interference is microns of axis length. Additionally, as shown previously, correlations between HEI10 focal intensity and foci

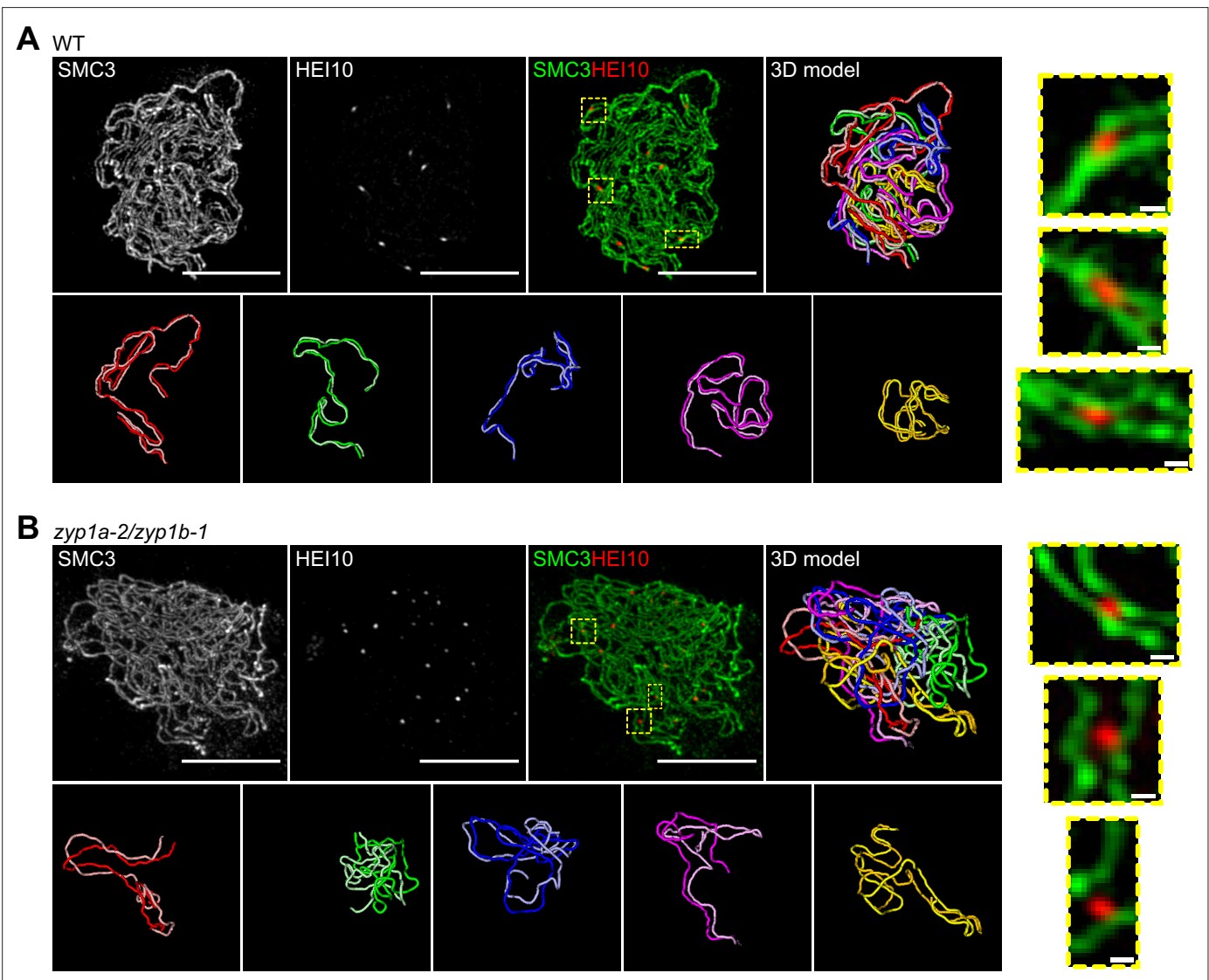

**Figure 3.** 3D-SIM imaging of late-prophase I cells. Maximum intensity projections of 3D image stacks from wild-type (**A**) and *zyp1a-2/zyp1b-1* mutants (**B**), labelled for SMC3 (green) and HEI10 (red) (scale bars = 5 µm). 3D models of segmented axial elements (with each chromosome axis labelled in a different colour), generated using the SNT plugin to FIJI, are also shown. Each nucleus contains five pairs of axes (bottom row, **A, B**), with individual axes in each pair having equivalent lengths and pairing tightly (**A**) or roughly (**B**) in 3D space. Zoomed-in regions (right-hand panels) from merged images (yellow dashed boxes) show the localisation of late-HEI10 foci between closely paired axes (scale bars = 0.2 µm).

The online version of this article includes the following figure supplement(s) for figure 3:

**Figure supplement 1.** 3D-SIM imaging of early pachytene cells.

number or position can be detected that provide strong experimental support for the coarsening model (*Morgan et al., 2021*).

We therefore used super-resolution microscopy and a bespoke image analysis pipeline to quantify the position and intensity of late-HEI10 foci (that are known to mark the positions of class I COs; *Chelysheva et al., 2012*), along late-prophase I chromosomes in wild-type and *zyp1a-2/zyp1b-1* SC null mutant lines (previously characterised in *France et al., 2021*; *Figure 3A and B*). Late-prophase I cells from wild-type and *zyp1a-2/zyp1b-1* plants were stained for the cohesin component SMC3 (*Lam et al., 2005*), and HEI10 (*Chelysheva et al., 2012*), which label the meiotic axis and putative CO sites, respectively, and imaged using 3D-SIM microscopy. In total, we imaged 10 cells from 4 wild-type plants and 14 cells from 4 *zyp1a-2/zyp1b-1* plants, which provided sufficient data to quantitatively compare with model simulations. By tracing along the linear SMC3 signals, using the SNT plugin to FIJI (*Arshadi et al., 2021*), we were able to segment the 10 axial elements in each cell (*Figure 3A and B*, top-right panels). In accordance with previous studies, we found that axial elements of homologous chromosomes were tightly juxtaposed along their entire lengths in wild-type cells, whilst in the *zyp1a-2/zyp1b-1* line the axes were only loosely paired along their lengths (*Figure 3A and B*, bottom rows), with the majority of HEI10 foci being clearly located between the two homologous axes at an equivalent position along their length (*Figure 3A and B*, right-hand boxes; *Capilla-Pérez et al., 2021*; *France et al., 2021*). Once the paths of segmented axes had been extracted from the images, we were then able to use an automated image analysis pipeline to assign HEI10 foci to specific positions along individual axes based on their local proximity to those regions ('Materials and methods'). Additionally, 3D-SIM analysis of early pachytene cells in both the wild-type and *zyp1a-2/zyp1b-1* lines revealed a greater number of less bright HEI10 foci than were observed in late-prophase I cells, which was consistent with expectations (*Figure 3—figure supplement 1*).

## Nucleoplasmic coarsening model successfully predicts HEI10 foci patterning and intensities in *zyp1* mutants

Using the experimental cytology and image analysis described above, we were able to construct cytological late-HEI10 foci maps from both wild-type and *zyp1a-2/zyp1b-1* SC null mutant plants (*Figure 4A*). In the wild-type, we found an average of 9.0 late-HEI10 foci per cell, while in the *zyp1a-2/zyp1b-1* line we found an average of 18.7 late-HEI10 foci per cell, which was broadly consistent with previous cytological analyses of CO numbers in similar lines (*Capilla-Pérez et al., 2021*; *France et al., 2021*).

When analysing the relative distribution of all late-HEI10 foci along chromosome pairs, we found that in the *zyp1a-2/zyp1b-1* line the distribution of late-HEI10 foci was shifted towards more distal positions, nearer the chromosome ends, when compared with the wild-type, which was also consistent with previous genetic analysis of *zyp1* null mutants (*Capilla-Pérez et al., 2021*; *Figure 4A*). Importantly, nucleoplasmic coarsening model fits were capable of recapitulating this distribution, with a preference for forming COs in more distal regions (*Figure 4A*). Such increases are a direct consequence of the additional initial HEI10 loading applied to RIs near chromosome ends (*Figure 1F*), which was a parameter that was included to improve the fit in the original coarsening model (*Morgan et al., 2021*). Without the inclusion of this parameter, we would expect a flat distribution with no preference for forming COs in more distal regions. Note that the extra loading here (up to 25% on the most distal 60% of each chromosome pair) is lower and more spread than that used in the original model (up to 100% on the most distal 20%) (*Figure 1—figure supplement 1*). The increased end loading of HEI10 was included in the original model because the dynamics of SC-mediated coarsening naturally decrease the likelihood of RIs close to chromosome ends forming COs. As there is an absence of SC-mediated coarsening in the nucleoplasmic coarsening model, each RI is equally capable of forming a CO regardless of its position. Thus, even though there is less pronounced HEI10 end loading in our nucleoplasmic coarsening model compared with our original model (25% vs. 100%), this lower end-bias still causes a relative increase in distally positioned COs under nucleoplasmic coarsening conditions compared with SC-mediated coarsening conditions. In the original coarsening model, we hypothesised this end bias results from preferential synapsis near the chromosome ends. In the absence of an SC, there is obviously no synapsis, which explains why the amount of end loading required within our nucleoplasmic-coarsening simulations is lower. However, the preferential pairing

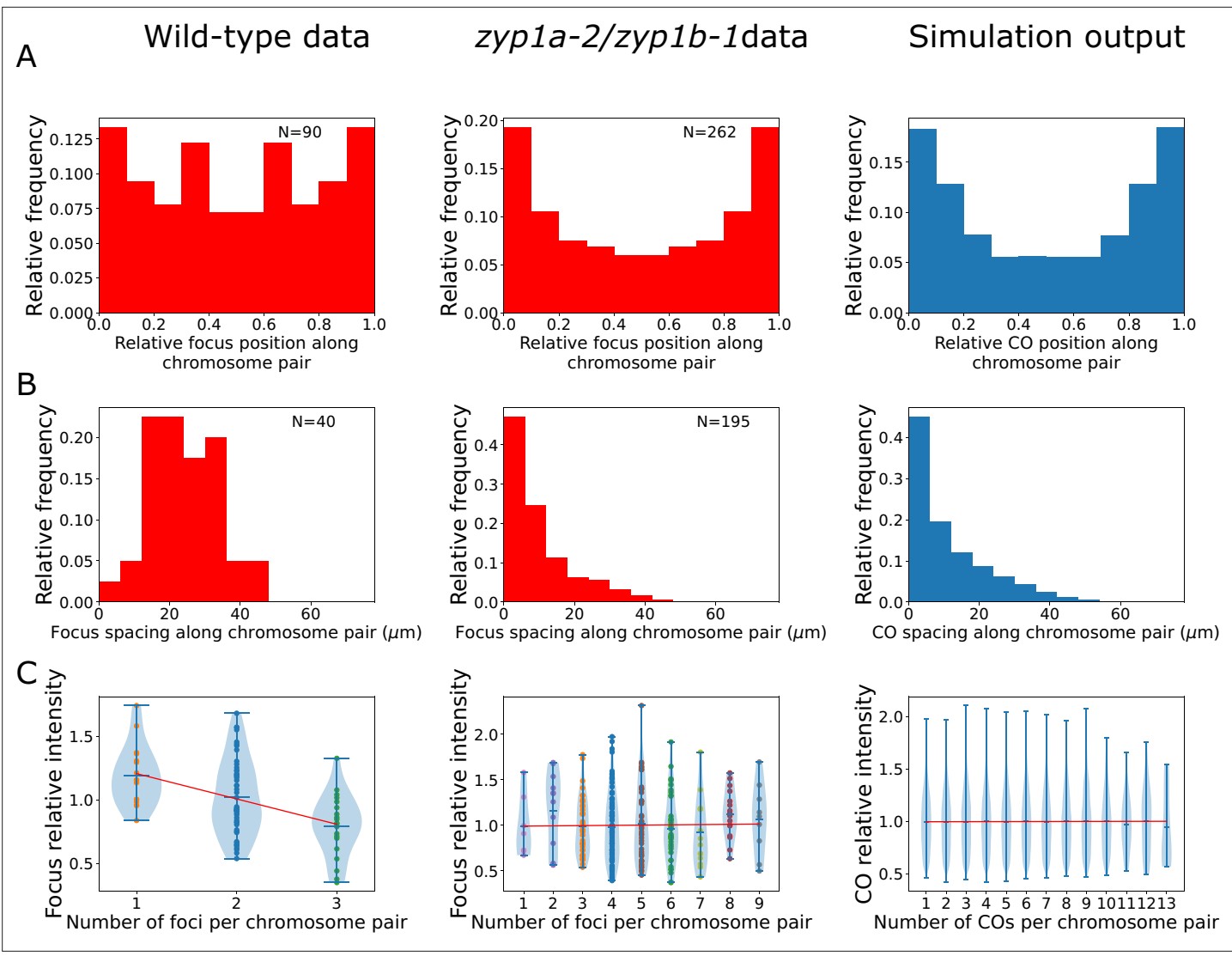

**Figure 4.** Analysis of late-HEI10 focus patterning and intensity data in wild-type and *zyp1* mutant. (**A–C**) Experimental data from wild-type plants (left), *zyp1a-2/zyp1b-1* plants (middle), and from nucleoplasmic coarsening model simulation outputs (right). Model outputs from simulating 10,000 cells are shown. (**A**) Late-HEI10 focus positions along chromosome pairs, relative to the length of the chromosome pair. Experimental data are replicated and made symmetric about chromosome midpoints (N = 161,322 simulated foci). (**B**) Distribution of spacing between adjacent late-HEI10 foci, in μm (N = 113,278 simulated spaces between foci). (**C**) Violin plots of late-HEI10 foci intensities against the number of late-HEI10 foci on that chromosome pair. Focus intensities are relative to the mean intensity of all (on-chromosome) HEI10 foci within the same cell. Red lines show simple linear regression best fits, treating the number of foci on the chromosome as a continuous independent variable. For wild-type plants, slope = –0.1944, $R^2$ = 0.188, $F_{(1,88)}$ = 20.36, p<0.001. For *zyp1* mutant plants, slope = 0.0028, $R^2$ < 0.001, $F_{(1,260)}$ = 0.04538, p=0.831. For simulated data, slope = 0.0004, $R^2$ < 0.001, $F_{(1,161320)}$ = 0.7838, p=0.376. Simulation parameters are again as listed in *Figure 1E*, with simulated data from 10,000 cells (50,000 chromosome pairs).

The online version of this article includes the following source data and figure supplement(s) for figure 4:

**Figure supplement 1.** A combined synaptonemal complex (SC)- and nucleoplasm-mediated coarsening model.

**Figure supplement 1—source data 1.** Default simulation parameter values for various scenarios: WT, original synaptonemal complex (SC)-mediated coarsening model with wild-type parameters (as implemented in 'Materials and methods'), WT+nuc, combined SC- and nucleoplasm-mediated coarsening model with wild-type parameters.

**Figure supplement 2.** The combined synaptonemal complex (SC)- and nucleoplasm-mediated coarsening model can explain crossover patterning in wild-type and HEI10 over-expressor lines.

of telomeres (to form the meiotic bouquet) (*Varas et al., 2015*), which presumably still happens in the absence of an SC, may explain why a smaller end bias for HEI10 loading persists.

We also examined the spacing between adjacent late-HEI10 foci on the same chromosome pair. In the wild-type, we found a scarcity of closely spaced crossovers, with a greater frequency of distantly spaced COs, reflecting the action of CO interference (*Figure 4B*). This was consistent with our previous analysis of wild-type CO patterning (albeit noisier due to the smaller sample size in this study) (*Morgan et al., 2021*). In the *zyp1a-2/zyp1b-1* line, however, we found a very high frequency of closely spaced crossovers, with a diminishing number of distantly spaced COs (*Figure 4B*). This behaviour reflects an absence of CO interference and provides strong cytological support for previous conclusions drawn from genetic experiments in *zyp1* null mutants (*Capilla-Pérez et al., 2021*). In this instance, the nucleoplasmic coarsening model was fully capable of predicting this pattern of closely spaced COs, without additional parameterisation, validating the model and reflecting an absence of CO interference (*Figure 4B*). Note that we use the term 'predicting' here and below to define instances where the model simulations were capable of explaining data that was not used to explicitly fit the model.

In addition to the late-HEI10 focus positioning data, we were also able to extract fluorescence intensity measurements from each HEI10 focus in our imaging data. In wild-type *Arabidopsis,* we previously showed that the normalised intensity of individual late HEI10 foci is negatively correlated with the number of foci per bivalent, which is consistent with an SC-mediated coarsening process (*Morgan et al., 2021*). Here, we again identified such a correlation in wild-type plants (*Figure 4C*). However, we did not find such a negative correlation in *zyp1a-2/zyp1b-1* mutants (*Figure 4C*). This would be expected from a nucleoplasmic coarsening model as, in this scenario, HEI10 is equally capable of diffusing through the nucleoplasm from any one RI to any other RI, regardless of whether the other RI is located on the same chromosome pair or a different chromosome pair (see *Figure 1A and B*). Indeed, simulations of the nucleoplasmic coarsening model again predicted these findings without the need for additional parameterisation and produced comparable results to the *zyp1a-2/zyp1b-1* line (*Figure 4C*).

## Combining synaptonemal complex- and nucleoplasm-mediated coarsening models

Although it appears likely that the SC acts as a conduit to promote HEI10 diffusion (*Morgan et al., 2021*; *Rog et al., 2017*; *Zhang et al., 2021*; *Zhang et al., 2018*), it remains unclear whether, in wild-type cells, the SC is capable of irreversibly compartmentalising the diffusion of HEI10, such that there is little or no recycling of SC-bound HEI10 back into the nucleoplasm. To address this question, we sought to combine our original SC-mediated coarsening model, which assumes there is no recycling of SC-bound HEI10 molecules back into the nucleoplasm (*Morgan et al., 2021*), with the nucleoplasmic coarsening model ('Materials and methods', *Figure 4—figure supplement 1*). We then tested whether this combined model was still capable of robustly reproducing the experimental CO patterning results we previously obtained from wild-type and HEI10 over-expressing *Arabidopsis* (*Morgan et al., 2021*; *Figure 4—figure supplement 2*).

We used a combined version of the coarsening model, where the rates of absorption and escape between the RI and the SC were set to be 90% of those in the original SC-mediated coarsening model (*Morgan et al., 2021*), and the exchange rates between the RI and nucleoplasm were set to be 10% of those in the above nucleoplasmic coarsening model. We found that this combined model was still fully capable of explaining the CO number and CO spacing distributions previously observed in wild-type *Arabidopsis* (*Figure 4—figure supplement 2*). Importantly, with this combined coarsening ratio, we still observed only a very small proportion (<0.2%) of bivalents with zero COs forming in our simulations and, hence, CO assurance is retained. This analysis demonstrates that a scenario where the SC preferentially promotes retention and diffusion of HEI10 along the SC, but without completely blocking HEI10 exchange between the SC and nucleoplasm, is still fully compatible with the coarsening model for CO patterning. Additionally, by increasing total HEI10 concentration (both along the SC and at RIs) within our combined coarsening model simulations by 4.5-fold, we were able the reproduce CO number and CO spacing results previously obtained from HEI10 over-expressor lines (*Figure 4—figure supplement 2*; *Morgan et al., 2021*).

## The combined coarsening model explains CO patterning in mutants with partial synapsis

As well as investigating coarsening in wild-type plants, HEI10 over-expressors and in mutants that completely lack an SC, we also sought to determine whether the combined coarsening model could explain CO patterning in mutants that exhibit partial, but incomplete, synapsis. In *Arabidopsis*, closely spaced crossovers have been observed in a number of mutants that exhibit partial synapsis, such as *axr1*, *pss1*, and *asy1* (*Duroc et al., 2014*; *Jahns et al., 2014*; *Lambing et al., 2020*; *Pochon et al., 2022*). Intuitively, this common observation could be explained by the coarsening model. In the presence of partial synapsis, the HEI10 protein would still preferentially load and diffuse along the SC, similar to the wild-type situation. However, as the total length of SC available for HEI10 loading would be less than wild-type, but the total nuclear concentration of HEI10 protein would presumably be the same as wild-type, we anticipate that a higher concentration of HEI10 would be loaded per micron of SC length. This situation is then similar to that found in a HEI10 over-expressor line, which has a greater frequency of closely-spaced COs than wild-type, as previously explained by the coarsening model (*Morgan et al., 2021*).

To test this theory, we investigated the patterning of late-HEI10 foci in an *Arabidopsis pch2* mutant. PCH2 is a conserved AAA+ATPase, with many known meiotic functions (*Bhalla, 2023*; *Lambing et al., 2015*; *Yang et al., 2022*; *Yang et al., 2020*). In *Arabidopsis*, loss of PCH2 has been shown to compromise SC polymerisation, generating a greater number of closely spaced COs, with no known effect on meiotic DSB number (*Lambing et al., 2015*). Once again, we used 3D-SIM microscopy to assess and quantitatively evaluate the positions of HEI10 foci in late-pachytene *pch2-1* mutants (*Figure 5A*). In total, we imaged 39 cells from three *pch2-1* plants. We found that late-HEI10 foci in *pch2-1* mutants were exclusively associated with the short SC segments that form. On average, *pch2-1* cells contained 14 (std. dev. ± 3) SC segments with an average length of 5.1 (std. dev. ± 4.4) µm. We therefore mapped the position of HEI10 foci along these short SC segments using our bespoke image analysis pipeline and quantified the number and position of late-HEI10 per SC segment within each nucleus (*Figure 5B–F*, left-hand plots). We detected 11.1 ($\sigma^2$ = 3.2) late-HEI10 foci per cell (*Figure 5B*), with an average of 0.79 ($\sigma^2$ = 0.48) late-HEI10 foci per SC segment (*Figure 5C*). We also found that all late-HEI10 foci were reasonably evenly distributed along SC segments (*Figure 5D*) and closely spaced late-HEI10 foci were frequently detected (*Figure 5E*). Also, the longer SC segments tended to have more late-HEI10 foci (*Figure 5F*), and the relative intensity of single late-HEI10 foci (i.e. foci on SC segments that possess only a single focus) was positively correlated with SC segment length (*Figure 5G*). The maximum length of an SC segment with zero late-HEI10 foci within our experimental sample was 9.6 µm, suggesting that ~10 µm is the minimum SC segment length required for CO assurance in *pch2-1* mutants.

To determine whether the combined coarsening model was capable of recapitulating the patterning effects observed in *pch2-1* mutants, we ran simulations using the same number and length distribution of SC segments per cell as in *pch2-1* mutants ('Materials and methods'). This arrangement differed from wild-type simulations where only five SC segments (corresponding to the five fully synapsed bivalents) were simulated for each cell. Thus, each SC segment was effectively treated as a discrete bivalent within our simulations (*Figure 5—figure supplement 1*). As no HEI10 foci were detected outside SC segments, no RIs were placed there, and these parts of the chromosomes were not included in our simulations.

In *pch2-1* simulations, the total length of the SC per cell is shorter (roughly one-quarter in length) than in wild-type, but the HEI10 amount is equivalent to that in wild-type simulations. This resulted in an overall higher (roughly four times) concentration of HEI10 per µm of SC and at RIs. This is akin to the situation in our HEI10 over-expressor simulations with SC-mediated coarsening (*Morgan et al., 2021*). However, as with HEI10 over-expressor simulations, this approximately fourfold increase in HEI10 concentration per µm of SC led to a smaller relative increase in the number of COs per unit length because SC-mediated coarsening dynamics proceed more rapidly when the RIs with high levels of HEI10 are closer together, as diffusion between them is easier. More rapid dynamics lead to relatively fewer COs, as coarsening proceeds further in the same time period, with HEI10 accumulating in a smaller number of bright foci. We therefore reduced the rates of HEI10 exchange between the RI and SC in order to slow the coarsening process, with an uptake rate of 20% and an escape rate of 15% of the rates in the original SC-mediated coarsening model. Changes to SC structure in the

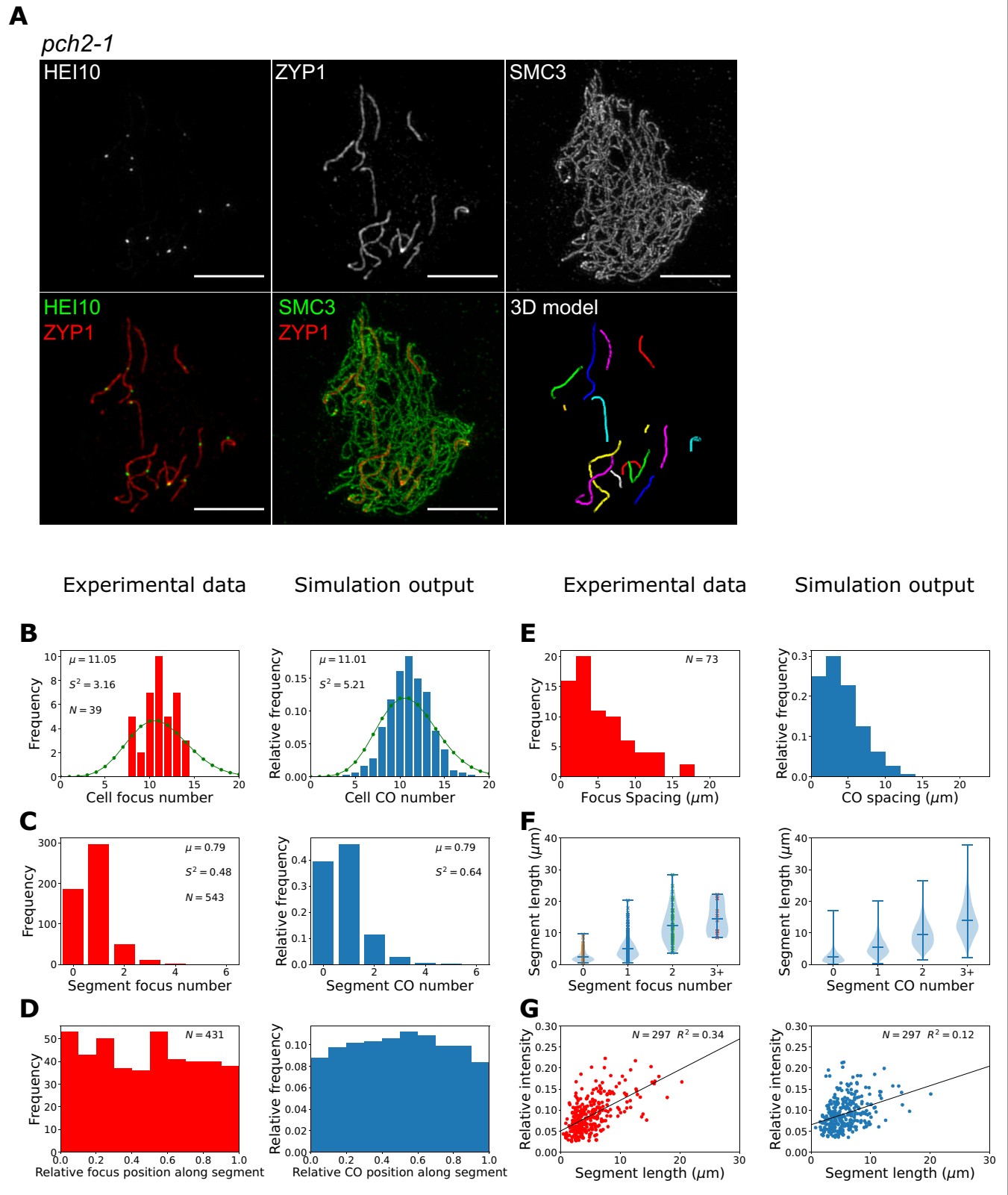

**Figure 5.** Late-HEI10 focus patterning and intensity data in *pch2-1*. (**A**) Maximum intensity projections of 3D image stacks from *pch2-1* mutants labelled for HEI10, ZYP1, and SMC3. A 3D model of segmented synaptonemal complex (SC) segments (with each segment labelled in a different colour), generated using the SNT plugin to FIJI, is also shown. (**B–G**) Experimental data and combined coarsening model simulations showing; (**B**) late-HEI10 focus number per cell (1000 simulated cells), green dots (joined by a line) show a Poisson distribution with the same mean, (**C**) late-HEI10 focus number

*Figure 5 continued on next page*

*Figure 5 continued*

per SC segment (13870 simulated segments), (**B, C**) sample mean (μ) and estimated variance ($S^2$) inset, (**D**) positioning of all late-HEI10 foci along SC segments relative to total segment length (11,009 simulated foci), (**E**) spacing between neighbouring late-HEI10 foci on the same SC segment (2590 spaces between simulated foci), (**F**) SC segment length versus number of late-HEI10 foci per SC segment (same number of observations as in **C**), and (**G**) single late-HEI10 focus intensity (relative to the mean intensity of all HEI10 foci within the same cell) versus SC segment length, with a random sample of simulation output to match experimental dataset size.

The online version of this article includes the following source data and figure supplement(s) for figure 5:

**Figure supplement 1.** Model for *pch2* simulations.

**Figure supplement 1—source data 1.** Parameters for *pch2* simulations that are different from those in *Figure 4—figure supplement 1—source data 1*.

synaptic segments that occur in *pch2* mutants, versus wild-type SC, could account for these modified exchange rates. These changes were combined with nucleoplasmic recycling at a rate of 10% of that in the nucleoplasmic coarsening model. These simulations were then able to qualitatively reproduce the patterning effects observed in our experimental data (*Figure 5B–G*, right-hand plots), although we note that agreement between experimental data and model outputs was slightly less good for *pch2-1* than for the *zyp1* mutant. Specifically, using the *pch2-1* parameter values we could generate an average of 11.0 (σ² = 5.2) COs per cell and 0.8 (σ² = 0.6) COs per SC segment, which is similar to the mean and variance values in our imaging data (*Figure 5B and C*). We also found that about 40% of simulated segments had no COs, which is similar to the proportion in our experimental data. In our simulations, zero CO segments occur as a consequence of segments lacking any RIs (occurring for about 22% of simulated segments) or due to insufficient HEI10 amounts being present at RIs on those SC segments at the end of the simulations (18% of simulated segments) either because of low initial HEI10 loading and/or HEI10 loss via nucleoplasmic recycling. Crossover assurance was retained for longer segments, with ~99.5% of segments longer than 10 μm having one or more crossover. Additionally, we recapitulated the relatively flat distribution of COs within SC segments (albeit with a slight peak in more central positions in simulation outputs that is not present in the experimental data, *Figure 5D*), as well as reproducing the spacing between adjacent COs (*Figure 5E*), and the positive correlations between both CO number (*Figure 5F*) and single late-HEI10 focus relative intensity (*Figure 5G*) and SC length.

## Discussion

It was recently shown that the SC, a highly conserved and prominent feature of meiotic prophase I, is required to promote CO interference and CO assurance in *A. thaliana* (*Capilla-Pérez et al., 2021*; *France et al., 2021*). However, the mechanistic details of precisely how the SC plays a role in mediating interference remained obscure. Here, using a combination of modelling and super-resolution microscopy, we have shown that CO patterning in the absence of an SC is consistent with a coarsening model for CO interference. Previously, we introduced a mechanistic, mathematical coarsening model that could explain CO patterning in wild-type *Arabidopsis*, as well as in HEI10 over- and under-expressor lines (*Morgan et al., 2021*). Here, we make just one major change to the model for SC mutants: that HEI10 can now exchange through a nucleoplasmic pool rather than being restricted to individual chromosomes, as in the wild-type. With this biologically plausible modification, the coarsening model is now also capable of explaining why *zyp1* mutants lack CO assurance and CO interference in *Arabidopsis*. Additionally, the model successfully predicts previously unexplored observations in *zyp1* SC mutants, including the pattern of closely spaced COs and the absence of an anticorrelation between HEI10 focal intensity and focus number per chromosome pair. A similar function of the SC, in preventing the diffusion of recombination proteins via the nucleoplasm, has been proposed in *C. elegans* (*Rog et al., 2017*; *Zhang et al., 2021*; *Zhang et al., 2018*). However, unlike in *C. elegans* where successful CO formation depends upon SC formation (*Colaiácovo et al., 2003*), in *Arabidopsis* SC formation and CO formation can be uncoupled (*Capilla-Pérez et al., 2021*; *France et al., 2021*), making *Arabidopsis* an excellent system for demonstrating the effects of nucleoplasmic coarsening. Overall, our work highlights the critical role of the SC in controlling the spatial compartment through which HEI10 diffuses.

Recently, a study in *Arabidopsis* also successfully utilised the mathematical coarsening model (originally derived in *Morgan et al., 2021*) to explain genetic CO patterning data from wild-type and HEI10 over-expressing *Arabidopsis* (*Durand et al., 2022*). The authors also hypothesised that nucleoplasmic coarsening could explain CO patterning in *zyp1* mutants, and in *zyp1* mutant and HEI10 over-expressing lines that form extremely high numbers of COs. However, unlike in this work, the dynamics of nucleoplasmic coarsening were not explicitly modelled (*Durand et al., 2022*).

Interestingly, while *Arabidopsis zyp1* mutants lose CO interference, they still display a non-random (sub-Poissonian) distribution of COs per cell. In wild-type cells, CO interference operating along individual chromosomes regulates total CO numbers along each chromosome. In turn, the combined effect of interference operating along all chromosome pairs within a cell causes global regulation of the total number of COs within a cell, with the variance in total CO number per cell being close to the sum of the per chromosome variances. In some species, COs have been shown to co-vary, meaning a high CO number on one chromosome generally corresponds with high CO numbers on other chromosomes within the same cell. This leads to a broader distribution of CO numbers per cell than might be expected (*Wang et al., 2019*). However, recent results suggest that CO co-variation is not present in *Arabidopsis* (*Durand et al., 2022*). *zyp1* mutants lose CO interference but retain per-cell constraints on total CO numbers, as evidenced by the Poissonian and sub-Poissonian distributions of total CO numbers on individual chromosome pairs and in individual cells, respectively. In other words, in the *zyp1* mutant, COs designated on different RIs are spatially independent from one another but their numbers are not independent. As noted in *Crismani et al., 2021*, an explanation for why CO numbers are relatively low (but still higher that wild-type) in SC mutants could be due to the influence of a limiting trans-acting factor, such as HEI10. Similarly, in our modified coarsening model, the limited nucleoplasmic pool of HEI10 explains why the distribution of total CO numbers per cell remains sub-Poissonian. Importantly, caution should be exercised when interpreting sub-Poissonian or Poissonian distributions of total CO number per cell as signs of functional or non-functional CO interference, respectively. As we show here, a sub-Poissonian distribution is perfectly compatible with non-functional CO interference.

Importantly, as well as explaining CO patterning in mutants without synapsis, we have shown that a version of the coarsening model that incorporates both nucleoplasmic and SC-mediated coarsening is still fully capable of explaining CO patterning in wild-type and HEI10 over-expressing *Arabidopsis*. This builds on a previous version of the coarsening model, where HEI10 diffusion was restricted exclusively to the SC (*Morgan et al., 2021*). We anticipate that this new, combined coarsening model likely better reflects the biological reality of the coarsening process, with the SC promoting and enhancing the diffusion of HEI10 along individual bivalents but with some exchange of HEI10 molecules between the SC and the nucleoplasm. This leakage is incorporated here as exchange between RIs and the nucleoplasm, but it is straightforward to also include direct exchange between the SC and the nucleoplasm in the framework of this model.

The complex interplay between SC-mediated and nucleoplasmic coarsening can also provide a potential mechanistic explanation for other crossover phenomena, such as the interchromosomal effect (*Miller, 2020*; *Termolino et al., 2019*). The interchromosomal effect describes the observation that when one pair of homologs experiences a reduction in crossover frequency (due to a structural variation, such as an inversion) this is accompanied by a corresponding increase in crossover frequency on the other, structurally normal, homolog pairs (*Miller, 2020*; *Termolino et al., 2019*). In this scenario, RIs may be inhibited from occurring normally at the sites of structural variations, leading to a lower level of HEI10 in those regions. In turn, this could lead to a redistribution of the limited pool of HEI10 to the other chromosomes, thus increasing the likelihood of those chromosomes receiving extra COs.

Additionally, we have shown that HEI10 coarsening can explain aspects of CO patterning in *pch2* mutants that exhibit partial, but incomplete, synapsis (*Lambing et al., 2015*). In this scenario, available HEI10 molecules could still load onto the SC and undergo coarsening, similar to the situation in the wild-type. However, if a wild-type level of HEI10 protein is loaded onto shorter stretches of SC, this results in an increased concentration of HEI10 per micron of SC and, akin to the situation in an otherwise wild-type HEI10 over-expressor (*Morgan et al., 2021*), an increase in closely spaced COs within those stretches. An abundance of closely spaced COs have also been reported in other *Arabidopsis* mutants that exhibit partial synapsis, such as *axr1*, *pss1*, and *asy1* (*Duroc et al., 2014*; *Jahns*

*et al., 2014*; *Lambing et al., 2020*; *Pochon et al., 2022*), suggesting that the coarsening model could explain CO patterning in a variety of synaptic mutants.

Together, the results we present here further reinforce the ability of the coarsening paradigm to explain numerous aspects of CO control in both wild-type and mutant *Arabidopsis* lines. These aspects include CO placement, CO frequency, CO assurance, CO interference, CO homeostasis, heterochiasmy, meiotic duration, interchromosomal effects, and diverse HEI10 focal intensity measurements. Given the conserved nature of HEI10 (*Agarwal and Roeder, 2000*; *Chelysheva et al., 2012*; *Jantsch et al., 2004*; *Lake et al., 2015*; *Qiao et al., 2014*; *Reynolds et al., 2013*), and the discovery of similar coarsening mechanisms in *C. elegans* (*Zhang et al., 2021*), it is likely that similar mechanisms regulate these same crossover patterning phenomena in a wide variety of sexually reproducing eukaryotes.

# Materials and methods

**Key resources table**

| Reagent type (species) or resource | Designation | Source or reference | Identifiers | Additional information |
|---|---|---|---|---|
| Gene (*Arabidopsis*) | ZYP1A | TAIR | AT1G22260 | |
| Gene (*Arabidopsis*) | ZYP1B | TAIR | AT1G22275 | |
| Gene (*Arabidopsis*) | PCH2 | TAIR | AT4G24710 | |
| Strain, strain background (*Arabidopsis*) | zyp1a-2/zyp1b-1 | *France et al., 2021* | | Supplied by Dr James Higgins, University of Leicester |
| Strain, strain background (*Arabidopsis*) | pch2-1 | Syngenta Arabidopsis Insertion Library (SAIL) | SAIL_1187_C06 | |
| Antibody | Anti-HEI10 (rabbit polyclonal) | *Lambing et al., 2015* | | Supplied by Prof Chris Franklin, University of Birmingham (1:500 dilution) |
| Antibody | Anti-SMC3 (rat polyclonal) | *Ferdous et al., 2012* | | Supplied by Prof Chris Franklin, University of Birmingham (1:500 dilution) |
| Antibody | Anti-ZYP1 (guinea-pig polyclonal) | *France et al., 2021* | | Supplied by Prof Chris Franklin, University of Birmingham (1:500 dilution) |
| Antibody | Alexa Fluor 555 goat anti-rabbit (goat polyclonal) | Thermo Fisher | RRID:AB_2535849 | (1:200 dilution) |
| Antibody | Alexa Fluor plus 488 goat anti-rat (goat polyclonal) | Thermo Fisher | RRID:AB_2896330 | (1:200 dilution) |
| Antibody | Alexa Fluor 647 goat anti-guinea-pig (goat polyclonal) | Thermo Fisher | RRID:AB_2735091 | (1:200 dilution) |
| Sequence-based reagent | PCH2_1_FV | *Lambing et al., 2015* | PCR primers | CAGTGCAAATAGCCGTCGCTGAG |
| Sequence-based reagent | PCH2_1_RV | *Lambing et al., 2015* | PCR primers | CTCACATGGTCCTTCTTCAATGAGC |
| Sequence-based reagent | Sail LB2 | *Lambing et al., 2015* | PCR primers | GCTTCCTATTATATCTTCCCAAATTACCAATACA |
| Sequence-based reagent | zyp1_ns_1 | *France et al., 2021* | PCR primers | CTCGCATTTGCTGGTTTAAAGAGTC |
| Sequence-based reagent | zyp1b_sp_1 | *France et al., 2021* | PCR primers | TGCGTATATTGCTAGGTTTATATTG |
| Sequence-based reagent | salk_lb2 | *France et al., 2021* | PCR primers | GTGCTTTACGGCACCTCGAC |
| Sequence-based reagent | zyp1a_sp_1 | *France et al., 2021* | PCR primers | GAATAGTTAGCAGATTCATATTTCAC |

*Continued on next page*

*Continued*

| Reagent type (species) or resource | Designation | Source or reference | Identifiers | Additional information |
|---|---|---|---|---|
| Peptide, recombinant protein | HindIII-HF | NEB | R3104S | |
| Chemical compound, drug | Cytohelicase | Sigma-Aldrich | C8274 | |
| Chemical compound, drug | Polyvinylpyrrolidone | Sigma-Aldrich | PVP40 | |
| Software, algorithm | FIJI | *Schindelin et al., 2012* | 2.1.0/1.53f51 | |
| Software, algorithm | Zen Black | Zeiss | 14.0.12.201 | |
| Software, algorithm | Python | Python | RRID:SCR_008394 | https://www.python.org/ |
| Software, algorithm | R | R Project for Statistical Computing | RRID:SCR_001905 | http://www.r-project.org/ |
| Software, algorithm | Julia | Julia | RRID:SCR_021666 | https://julialang.org/ |

## Plant materials

*A. thaliana* lines used in this study were wild-type Col-0, *pch2-1* (SAIL_1187_C06) and the *zyp1a-2/zyp1b-1* null-mutant (previously characterised in *France et al., 2021*). The *zyp1a-2/zyp1b-1* line contains a CRISPR/Cas9 derived 14 base-pair deletion in exon 3 of *ZYP1A* and a T-DNA insertion in exon 3 of *ZYP1B*. Plants were grown in controlled environment rooms with 16 hr of light (125 mMol cool white) at 20°C and 8 hr of darkness at 16°C.

## Immunocytology

Immunocytology of spread *A. thaliana* pachytene cells was performed as described previously (*Morgan et al., 2021*; *Morgan and Wegel, 2020*). In brief, staged anthers were dissected from *Arabidopsis* floral buds and macerated in a 10 µl drop of digestion medium (0.4% cytohelicase, 1.5% sucrose, 1% polyvinylpyrolidone in sterile water) on a No. 1.5H coverslip (Marienfeld). Coverslips were then incubated in a moist chamber at 37°C for 4 min before adding 10 µl of 2% lipsol solution followed by 20 µl of 4% paraformaldehyde (pH 8). Coverslips were dried for 3 hr in the fumehood and then blocked in blocking buffer (0.3% BSA in 1× PBS) for 15 min. 50 µl of primary antibody, diluted in blocking buffer, was added to the coverslips and they were then incubated overnight in a moist chamber at 4°C. 50 µl of secondary antibody, diluted in blocking buffer, was added to the coverslips, and they were then incubated in a moist chamber at 37°C for 2 hr. Coverslips were then incubated in 10 µl DAPI (10 µg/ml) for 5 min before adding 7 µl of Vectashield and mounting them on a glass slide. Coverslips were washed for 3 × 5 min in 1× PBS 0.1% Triton before and after each incubation step. The following primary antibodies were used at 1:500 dilutions: anti-SMC3 (rat) (*Ferdous et al., 2012*), anti-HEI10 (rabbit) (*Lambing et al., 2015*), and anti-ZYP1 (guinea-pig) (*France et al., 2021*). The following secondary antibodies were used at 1:200 dilutions: anti-rat Alexa Fluor Plus 488 (Thermo Fisher A48262), anti-rabbit Alexa Fluor 555 (Thermo Fisher A21428), and anti-guinea-pig Alexa Fluor 647 (Thermo Fisher A21450). Immunostained cells were imaged using 3D structured illumination microscopy (3D-SIM) on a Zeiss Elyra PS1 microscope equipped with an EM-CCD camera, a Plan-Apochromat ×63, NA 1.40 oil objective and 405, 488, 561, and 642 nm solid-state laser diodes. Cells were imaged with three stripe angles and five phases and z-stacks were captured at an interval size of 0.0909 µm. An immersion oil with a refractive index of 1.515 was used that was optimised for the green/red (SMC3/HEI10) channels of our system. For optimal image quality and to minimise the introduction of reconstruction artefacts,

microscope laser power and camera gain values were manually adjusted for each cell. Bleaching and contrast of raw images was assessed using the SIMcheck plugin to FIJI (**Ball et al., 2015**).

## Image analysis

For wild-type and *zyp1a-2/zyp1b-1* nuclei, individual axes were traced and segmented in three-dimensions from 3D-SIM z-stacks using the SNT plugin to FIJI (**Arshadi et al., 2021**). Traces were made using the linear SMC3 signal, which localises along the meiotic axis, and 3D-skeletons of each axis-traced path were generated using SNT. For *zyp1a-2/zyp1b-1* nuclei, each axial element was traced end-to-end to generate 10 complete skeleton traces for each nucleus. For wild-type nuclei, due to the proximity of the two parallel SMC3 signals, a single trace was used to segment the paths of each pair of synapsed axial elements, generating five complete skeleton traces for each nucleus. Foci were detected within images using the FociPicker3D FIJI plugin (**Du et al., 2011**). Uniform background intensity thresholds and minimum pixel size values were manually optimised within a range of 6,000–20,000 and 5–20, respectively, with otherwise default parameters. Further analysis was performed using custom Python scripts. For *zyp1a-2/zyp1b-1* nuclei, foci were assigned to the chromosome pair for which the average of the two distances from the closest points on each of the axes was smallest, provided this was less than a threshold distance (0.5 µm). If this was not the case, then the foci was assigned to the closest chromosome pair, provided that it was nearer to one of the two axes than a smaller threshold (0.3 µm). Foci satisfying neither of these two criterion were considered as off target signal and ignored. For wild-type nuclei, each focus was assigned to the closest trace, provided that it was nearer than a threshold (0.5 µm). Again, foci not satisfying this criterion were considered as off target signal and ignored. Relative positions along chromosome pairs were calculated by dividing the distance along the traced axis (measured in terms of voxels) by the total length of the trace. Foci intensities were calculated using the mean intensity within a sphere of radius 0.1985 µm (five voxels in the x and y planes), centred at the foci peak, and the median intensity of the image (used as an estimate of the background fluorescence) was subtracted. All intensity values used were normalised by dividing the foci intensity by the mean of the intensities of all foci within the same cell (only considering foci associated with a chromosome pair). Simple linear regression was performed using the Python statsmodels package.

For *pch2-1* nuclei, segments of synapsis were traced and segmented in three-dimensions from 3D-SIM z-stacks using the SNT plugin to FIJI (**Arshadi et al., 2021**). Traces were made using the linear ZYP1 signal, which localises along the SC, and 3D skeletons of each SC-traced path were generated using SNT. Each SC segment was traced end-to-end to generate between 8 and 21 complete skeleton traces for each nucleus. Individual traces varied from 0.4 to 28.4 µm in length. Again, foci were detected within images using the FociPicker3D FIJI plugin (**Du et al., 2011**). Uniform background intensity thresholds and minimum pixel size values were manually optimised within a range of 10,000–15,000 and 10–20, respectively, with otherwise default parameters. Further analysis was performed using custom Python scripts. All subsequent analyses proceeded as for wild-type nuclei.

## Mathematical models

### Nucleoplasmic coarsening model

In each cell, $Q = 5$ chromosome pairs are simulated. The $q$th chromosome pair is taken to have a length $L_q$, sampled from the (per-chromosome) normal distribution $\mathcal{N}_T\left(\mu_q, \sigma_q^2, -3, 3\right)$ truncated at 3 standard deviations from the mean (to ensure positive chromosome lengths). We use the same values as in our earlier work (**Morgan et al., 2021**), fitted to experimentally measured wild-type chromosome length distributions: $\mu_1 = 37.5, \mu_2 = 40.4, \mu_3 = 45.7, \mu_4 = 53.8, \mu_5 = 59.8$, and $\sigma_1 = 4.93, \sigma_2 = 5.06, \sigma_3 = 5.86, \sigma_4 = 7.18, \sigma_5 = 7.13$, with all these parameters having units of $\mu$m. We chose to retain these original values as they were broadly consistent with the experimental chromosome lengths measured in this study, but were calculated from a greater sample size.

RIs are placed randomly along each of the chromosome pairs, with a probability density $\rho$ per µm. To implement this, for each simulated chromosome pair with length $L$, a random number is generated from a Poisson process with mean $\rho L$, and then this number of RIs is placed uniformly at random along that chromosome pair. We note that this procedure differs slightly from the method used in our original SC coarsening model, where floor$\left(\rho L\right)$ crossovers were placed on a chromosome with length $L$. Simulation outputs with this original rule (not shown) differ only slightly.

The pro-crossover factor HEI10 is present in a nucleoplasmic pool, with amount $P$, and at compartments at each of the RIs, with amounts $C_{q,n}$, $1 \leq q \leq Q = 5$, $1 \leq n \leq N_q$ for $n$th RI on the $q$th chromosome pair. HEI10 is able to move between the nucleoplasmic pool to each of the RI compartments at rate $\alpha$, and to escape from the RI compartments back into the pool at a rate $\beta(C_{q,n})$, which depends on the amount of HEI10 within each individual RI compartment. *Figure 1D* shows the functional form of this rate in more detail, from which we see that it is a strictly decreasing function of the RI HEI10 amount $C_{q,n}$. As in our SC interference model, we take

$$\beta(C_{q,n}) = \frac{\beta_C}{1 + (C_{q,n}/K_C)^{\gamma}}.$$

The system of ordinary differential equations governing HEI10 amounts is

$$\frac{dC_{q,n}}{dt} = \left(\alpha P - \beta(C_{q,n}) C_{q,n}\right) H_s\left(C_{q,n} - K_H\right), \quad 1 \leq q \leq Q, 1 \leq n \leq N_q,$$

$$\frac{dP}{dt} = \sum_{q=1}^{Q} \sum_{n=1}^{N_q} H_s\left(C_{q,n} - K_H\right) \left(\beta(C_{q,n}) C_{q,n} - \alpha P\right).$$

Compared to our earlier model (*Morgan et al., 2021*), we also incorporated a further modulation to the HEI10 exchange rates between the nucleoplasm and RIs through the multiplicative smoothing function $H_s$. For HEI10 levels in an RI well below a threshold $K_H$, the smoothing function becomes close to zero, while for levels well above, the function saturates to one, with a smooth crossover in between. This feature ensures that RIs with HE10 levels well below the threshold become disconnected, with their HEI10 effectively disappearing from the rest of the system. This reflects the experimental observation that smaller HEI10 foci are not detected in late-pachytene nuclei (*Morgan et al., 2021*), with RIs that lack sufficient HEI10 presumably being channelled into a non-crossover or class II repair pathway. We implement this through the smooth cutoff function $H_s$:

$$H_s(s) = \tfrac{1}{2}\left(1 + \tanh\left(K_s s\right)\right)$$

which multiplies the HEI10 exchange rates between the RIs and nucleoplasm. Overall, this system conserves the total amount of HEI10 within each cell (the nucleoplasmic pool plus all RI compartments), rather than within each SC, which was the case for our earlier SC-mediated coarsening model (*Morgan et al., 2021*).

Unlike in our previous SC-mediated coarsening model, where the total HEI10 amount in each cell depended on the total length of SC and the total number of RIs (which are relatively constant from cell to cell), we adopted an alternative approach for controlling cellular HEI10 amount in our simulations. This change was necessary due to the absence of an SC in the nucleoplasmic-coarsening model and is likely more realistic, with cellular HEI10 protein amount being controlled independently of SC formation, as we now describe.

The initial total amount of HEI10 within each cell, $F$, is sampled from a truncated normal distribution

$$F = \mathcal{N}_T\left(\mu_F, \sigma_F^2, -3, 3\right),$$

where the mean $\mu_F$ and standard deviation $\sigma_F$ are taken to be the mean and standard deviation of the total amounts of HEI10 within each cell in the original SC-mediated coarsening model (*Morgan et al., 2021*). This initial cellular level of HEI10 is divided between the RIs and the nucleoplasmic pool, at constant proportions $R_{RI}$ and $R_P$, respectively, with $R_{RI} + R_P = 1$.

The total amount of RI HEI10 is then randomly divided between each RI, where, for each RI, a random weighting $\hat{C}_{q,n}$ is generated from

$$\hat{C}_{q,n} = f\left(\frac{x_{q,n}}{L_q}\right) X_{q,n}, \quad X_{q,n} = \mathcal{N}_T\left(1, \sigma_C^2, -3, 3\right), \quad 1 \leq q \leq Q, 1 \leq n \leq N_q,$$

where $\mathcal{N}_T$ again denotes a truncated normal distribution and $x_{q,n}$ is the position of the $n$th RI on the $q$th chromosome pair. The piecewise function $f(x/L)$, included to account for the increased frequency of crossovers towards the end of each chromosome, is defined as

$$
f(\xi) = \begin{cases} 1 + (f_e - 1)\ (1 - \xi/x_e) & 0 \le \xi \le x_e, \\ 1 & x_e \le \xi \le 1 - x_e, \\ 1 + (f_e - 1)\ (\xi - 1 + x_e)\ /x_e & 1 - x_e \le \xi \le 1. \end{cases}
$$

The total initial amount of HEI10 at all RIs, $R_{RI}F$, is then loaded at the RIs in proportion to these weights, namely

$$
C_{q,n}(t = 0) = R_{RI}F \frac{\hat{C}_{q,n}}{\sum\limits_{q'=1}^{Q} \sum\limits_{n'=1}^{N_q} \hat{C}_{q',n'}}.
$$

The remaining HEI10 is placed in the nucleoplasmic pool, with initial amount $P(t = 0) = R_P F$, which has the same average as the average amount of HEI10 loaded onto all the SCs in the original SC-mediated coarsening model (**Morgan et al., 2021**).

This system is simulated for a duration $T = 10\text{hr}$, a duration of pachytene equivalent to that used in our original model (**Prusicki et al., 2019**). Numerical simulations are performed using the Rodas5 solver (**di Marzo, 1992**) of the DifferentialEquations.jl Julia package (**Rackauckas and Nie, 2017**), with a non-negativity constraint on HEI10 amounts. The initial conditions for the simulation are random, but the evolution of the system is deterministic. Multiple simulations (10,000) with different random initial conditions are performed to determine the distribution of crossover numbers, positions, and intensities. At the end of each simulation, RIs are designated as crossover sites according to a modified rule (see 'Modified CO criterion' below).

All parameters were chosen to be the same as those as in the original SC coarsening model (**Morgan et al., 2021**), except for the RI HEI10 absorption rate $\alpha$, and the (maximum) escape rate $\beta_C$, which was multiplied by a factor of 0.06 (**Figure 1E**). These changes reflect that HEI10 is now being exchanged with the nucleoplasmic pool rather than locally on the synaptonemal complex.

For simulations of the *zyp1* HEI10 over-expressor line, a 3.5-fold change in HEI10 levels was used, but a larger fold change was chosen for the standard deviation $\sigma_F$ of total cellular HEI10 amounts, in order to generate greater cell–cell variability, with an increase from $3.5 \times 75$ a.u. = 263 a.u. to 862 a.u. We also verified that increasing the noise to this degree in the original model of **Morgan et al., 2021** (as implemented below) for wild-type and HEI10 over-expressing lines did not make a substantial difference to model predictions for the number of COs per SC or the positioning of COs. We note that the above 3.5-fold change in HEI10 levels resulted in roughly the same relative change in CO number, unlike for SC-mediated coarsening where 4.5 times the level of HEI10 only resulted in approximate doubling of the number of COs (**Morgan et al., 2021**). This reduced change reflects the reduced sensitivity of the SC-mediated coarsening to changes in HEI10 levels compared to nucleoplasmic-mediated coarsening (for these choices of parameter values).

## SC-mediated coarsening model

This model is as described in our earlier work, but with the introduction of the smooth cutoff function preventing recycling at RIs with low levels of HEI10, with slightly different initial conditions, and with a new rule for determining which RIs are crossovers at the end of the simulation. Again, $Q = 5$ chromosomes are simulated in each cell, these having SCs with lengths $L_q$ generated randomly as in the nucleoplasmic coarsening model. HEI10 is present on each of the SCs with concentration $c_q(x, t)$ per unit length and is able to diffuse along them with (one-dimensional) diffusion coefficient $D$. RI numbers and positions are generated as in the nucleoplasmic model. Again, we note that this is slightly different to the rule used in **Morgan et al., 2021**, but that this change has limited effect on model outputs. The variables $C_{q,n}$ represent the amount of HEI10 associated with the $n$th RI on the $q$th SC. However, RIs are not able to exchange HEI10 with the nucleoplasm, but instead exchange HEI10 with the relevant SC, with uptake rate $\hat{\alpha}$ and escape rate $\hat{\beta}(C_{q,n})$, this escape rate taking the same functional form as in the nucleoplasmic coarsening model but with the new parameter $\hat{\beta}_C$. This gives the system of equations

$$
\frac{dC_{q,n}}{dt} = \left( \hat{\alpha} c_q(x_{q,n}) - \hat{\beta}(C_{q,n}) C_{q,n} \right) H_s(C_{q,n} - K_H), \quad 1 \le q \le Q, 1 \le n \le N_q,
$$

$$\frac{\partial c_q}{\partial t} = D\frac{\partial^2 c_q}{\partial x^2} + \sum_{n=1}^{N_q}\left(\hat{\beta}\left(C_{q,n}\right)C_{q,n} - \hat{\alpha}c_q\left(x_{q,n}\right)\right)H_s\left(C_{q,n}-K_H\right)\delta\left(x-x_{q,n}\right),$$

$$1 \le q \le Q,\ 0 \le x \le L_q,$$

$$\frac{\partial c_q}{\partial x} = 0,\ \text{at } x = 0 \text{ and } x = L_q,$$

$$c_q = c_0 \text{ at } t = 0.$$

Initial conditions for total cellular HEI10 levels are as in the nucleoplasmic model. Initial RI HEI10 amounts are as in the nucleoplasmic model, but with different parameters (see *Figure 4—figure supplement 1*) in the non-uniform loading function $f(\xi)$. However, instead of being placed in the nucleoplasmic pool, the remaining HEI10 is placed uniformly on the SCs, giving

$$c_q\left(t=0\right) = c_0 = \frac{R_{SC}F}{\sum\limits_{q'=1}^{Q}L_{q'}},$$

where $R_{SC} = 1 - R_{RI}$. Note that these initial conditions differ from those used in *Morgan et al., 2021*, but when the new rule for determining crossovers is used, they give very similar results in terms of crossover number and location. This model was used to produce the simulation output shown in *Figure 4—figure supplement 2* (WT, OX), using the parameters shown in *Figure 4—figure supplement 1D* (WT column), except for the over-expressor OX case, where the parameters $\mu_F$ and $\sigma_F$ controlling initial cellular HEI10 levels were increased to 4.5× their WT values.

## Combined coarsening model

The combined model includes all the above rules in the nucleoplasmic and SC-mediated coarsening models, with HEI10 being present in RI, SC and nucleoplasm compartments. The governing equations are a combination of those in the two other models; explicitly

$$\frac{dC_{q,n}}{dt} = \left(\alpha P - \beta\left(C_{q,n}\right)C_{q,n} + \hat{\alpha}c_q\left(x_{q,n}\right) - \hat{\beta}\left(C_{q,n}\right)C_{q,n}\right)H_s\left(C_{q,n}-K_H\right),$$

$$1 \le q \le Q,\quad 1 \le n \le N_q,$$

$$\frac{\partial c_q}{\partial t} = D\frac{\partial^2 c_q}{\partial x^2} + \sum_{n=1}^{N_q}(\hat{\beta}(C_{q,n})C_{q,n} - \hat{\alpha}\,c_q(x_{q,n}))H_s(C_{q,n}-K_H)\delta(x-x_{q,n}),$$

$$1 \le q \le Q,\quad 0 \le x \le L_q,$$

$$\frac{dP}{dt} = \sum_{q=1}^{Q}\sum_{n=1}^{N_q}\left(\beta\left(C_{q,n}\right)C_{q,n} - \alpha P\right)H_s\left(C_{q,n}-K_H\right).$$

Initial and boundary conditions for wild-type or HEI10 over-expressor plants are as in the SC-mediated coarsening model above. No HEI10 is placed initially in the nucleoplasm. Each SC is discretised into m = 2000 compartments, and the spatial derivatives approximated by finite differences. This model was used to generate the simulation output shown in *Figure 4—figure supplement 2* (WT+nuc, OX+nuc), using the parameters shown in *Figure 4—figure supplement 1D* (WT+nuc column), except for the over-expressor OX+nuc case, where the parameters $\mu_F$ and $\sigma_F$ controlling initial cellular HEI10 levels were increased to 4.5× their WT values.

For the *pch2-1* mutant, we have a variable number of SC segments within the cell, which are here treated as distinct SCs. The number of SC segments is drawn randomly from a Poisson distribution with mean 13.92 (close to that measured experimentally), while the length of each SC segment is drawn from a Gamma distribution with parameters $L_\alpha = 1.693$, $L_\theta = 2.984\,\mu$m, so having mean $L_\alpha L_\theta \approx 5.1\mu$m and variance $L_\alpha L_\theta^2 \approx 15.1\mu$m$^2$. This distribution has probability distribution function

$$f_\Gamma\left(x; L_\alpha, L_\theta\right) = \frac{x^{L_\alpha-1}e^{-x/L_\theta}}{\Gamma(L_\alpha)L_\theta^{L_\alpha}},\ x > 0,$$

where $\Gamma(x)$ is the Gamma function and was fitted to experimental measurements of SC segment length. This statistical model for SC segment lengths introduced substantially greater variation in total cellular SC length than in the other simulations, which motivated our new rule for the assignment of initial HEI10 amounts. Initial and boundary conditions are the same here as for the SC-mediated coarsening model above (although there are a larger number of SCs here).

Owing to this large number of SCs, all of which are coupled with each other through the nucleoplasm, for efficient simulation each SC segment is spatially discretised into $m = 200$ portions, fewer than the number ($m = 2000$) used for each SC in the SC-mediated coarsening model above.

The smooth cutoff function $H_s$ is included so that RIs with low levels of HEI10 are unable to interact with the nucleoplasmic pool. This is necessary as otherwise these RIs provide an additional route for the transfer of HEI10 between two RIs on the same SC: HEI10 can diffuse along the SC from an RI losing HEI10 to a nearby RI with low levels of HEI10, escape into the nucleoplasm, be reabsorbed at another RI with low levels of HE10, and then diffuse along the SC to an RI gaining HEI10. This route causes the dynamics to progress too quickly.

The parameterisation of this combined model used for the *pch2-1* mutant has substantially smaller values for the rates of exchange of HEI10 between RIs and SCs ($\hat{\alpha}$ and $\hat{\beta}_C$) than in the original SC-mediated coarsening model (*Figure 5—figure supplement 1C*). These were chosen to better fit the experimental data, which shows a less pronounced spatial pattern than the wild-type data. Nucleoplasmic recycling was also set at a rate of 10% of that in the nucleoplasmic coarsening model. This model was used to generate the simulation output shown in *Figure 5*.

## SC-mediated coarsening model analysis

In the SC-mediated coarsening model, as the timescale for diffusion along the SC ($\sim L^2/D$, which is at most (60 μm)$^2$/1.1 μm$^2$ s$^{-1}$ ≈ 3300 s) is small compared with the duration of pachytene (36,000 s), the HEI10 concentration adopts a piecewise linear profile along each SC, with discontinuities in its slope at each RI. When there are just two RIs with HEI10 amounts $C_1$ and $C_2$, separated by a distance $\Delta x = x_2 - x_1$, we have that

$$\tfrac{dC_1}{dt} = \hat{\alpha}c(x_1) - \hat{\beta}(C_1)C_1 = -J, \quad \tfrac{dC_2}{dt} = \hat{\alpha}c(x_2) - \hat{\beta}(C_2)C_2 = J,$$

where the net flux of HEI10 from $C_1$ to $C_2$, $J$, is given by

$$J = D\tfrac{c(x_1) - c(x_2)}{\Delta x}.$$

On eliminating $c(x_1)$ and $c(x_2)$, the concentrations of HEI10 on the SC at the two RIs, we find that

$$\tfrac{d}{dt}\left(C_2 - C_1\right) = \tfrac{\hat{\beta}(C_1)C_1 - \hat{\beta}(C_2)C_2}{1 + \frac{\hat{\alpha}\Delta x}{2D}}.$$

We can see from this equation that closely spaced pairs of RIs, with small $\Delta x$, coarsen more quickly than those further apart. This effect generates spatial patterns in CO positioning, even in the absence of non-uniform initial HEI10 loading. However, this difference in coarsening rate is only significant if $\frac{\hat{\alpha}\Delta x}{2D}$ is not small compared with 1. In the limit where $\frac{\hat{\alpha}\Delta x}{2D}$ is small, the spatial patterns in CO positioning vanish, other than those spatial biases from non-uniform initial HEI10 loading.

With $K_C = 1$ and for $C_1, C_2 \gg 1$ and $\frac{\hat{\alpha}\Delta x}{2D} \gg 1$, which is the case for our original SC-mediated coarsening parameters, this can be approximated further as

$$\tfrac{d}{dt}\left(C_2 - C_1\right) = \tfrac{2D\hat{\beta}_c}{\hat{\alpha}\Delta x}(C_1^{1-\gamma} - C_2^{1-\gamma}).$$

From this equation, we find that coarsening proceeds faster for larger diffusion coefficients, larger escape rates from RIs to the SC, and slower for larger uptake rates from the SC to RIs, more widely spaced RIs, and when HEI10 levels are greater (as $\gamma > 1$). When more than two RIs are participating in coarsening, a more complex system of equations is obtained, but again containing the model parameters in the same combination $\frac{D\hat{\beta}_c}{\hat{\alpha}}$. As a consequence of this, it is difficult from simulation outputs to independently identify the parameters $\hat{\alpha}$, $\hat{\beta}_C$, and $D$, with many parameter sets being potentially compatible with our original findings.

## Modified crossover criterion

In our original simulations (*Morgan et al., 2021*) of SC-mediated coarsening in wild-type *A. thaliana*, we used a per-SC criterion to decide which RIs at the end of the simulation should be designated as crossovers. This criterion was chosen to be a threshold of 40% of the maximum HEI10 amount at

those RIs on the same SC. However, this criterion needed to be changed to study nucleoplasmic-mediated coarsening, and a simple absolute threshold would require substantial reparameterisation of the HEI10 model.

Instead, we designed a new per-cell criterion. RIs were designated as crossovers if they contained an amount of HEI10 more than 60% of the mean amount of HEI10 at an RI in that cell, where this mean excluded RIs with less HEI10 than an absolute threshold of 1 a.u. A per-cell threshold based on the mean amount rather than the maximum amount was used because of the substantial variation in maximum HEI10 level found in simulations of the *pch-1* mutant. All simulation outputs in this article use this adjusted threshold. We found that this made only a limited change to the results of simulating SC-mediated coarsening in WT and HEI10 over-expressing cells.

## Acknowledgements

We gratefully acknowledge James Higgins for providing the *zyp1a-2/zyp1b-1* mutant seeds, Chris Franklin for supplying the SMC3, HEI10, and ZYP1 antibodies, Eva Wegel for microscopy support, and members of the M Howard and X Feng labs for fruitful discussions on this topic. This work was supported by a BBSRC Discovery Fellowship (BB/V005774/1) awarded to CM and by BBSRC Institute Strategic Programme GEN (BB/P013511/1) to MH.

## Additional information

### Funding

| Funder | Grant reference number | Author |
|---|---|---|
| Biotechnology and Biological Sciences Research Council | BB/V005774/1 | Chris Morgan |
| Biotechnology and Biological Sciences Research Council | BB/P013511/1 | Martin Howard |

The funders had no role in study design, data collection and interpretation, or the decision to submit the work for publication.

### Author contributions

John A Fozard, Conceptualization, Data curation, Software, Formal analysis, Investigation, Methodology, Writing – original draft, Writing – review and editing; Chris Morgan, Conceptualization, Data curation, Formal analysis, Funding acquisition, Investigation, Methodology, Writing – original draft, Writing – review and editing; Martin Howard, Conceptualization, Supervision, Funding acquisition, Methodology, Writing – original draft, Project administration, Writing – review and editing

### Author ORCIDs

John A Fozard ⓘ http://orcid.org/0000-0001-9181-8083
Chris Morgan ⓘ http://orcid.org/0000-0002-7475-2155
Martin Howard ⓘ http://orcid.org/0000-0001-7670-0781

### Decision letter and Author response

Decision letter https://doi.org/10.7554/eLife.79408.sa1
Author response https://doi.org/10.7554/eLife.79408.sa2

## Additional files

### Supplementary files
• MDAR checklist

## Data availability

Imaging data associated with this study are available at https://doi.org/10.6084/m9.figshare.19650249.v1, https://doi.org/10.6084/m9.figshare.19665810.v1 and https://doi.org/10.6084/m9.figshare.21989732. Custom Python scripts for data analysis are available at https://github.com/jfozard/hei10_zyp1, (copy archived at *Fozard, 2023*).

The following datasets were generated:

| Author(s) | Year | Dataset title | Dataset URL | Database and Identifier |
|---|---|---|---|---|
| Fozard JA, Morgan C, Howard M | 2022 | ZYP1 data | https://doi.org/10.6084/m9.figshare.19650249.v1 | figshare, 10.6084/m9.figshare.19650249.v1 |
| Fozard JA, Morgan C, Howard M | 2022 | Col-o (for zyp1 comparison) | https://doi.org/10.6084/m9.figshare.19665810.v1 | figshare, 10.6084/m9.figshare.19665810.v1 |
| Fozard JA, Morgan C, Howard M | 2023 | pch2 data | https://doi.org/10.6084/m9.figshare.21989732 | figshare, 10.6084/m9.figshare.21989732 |

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
