## [Editor Report]

This important paper discloses a new control mechanism of meiotic crossing over, which is essential for the segregation of homologous chromosomes. With mathematical modeling and super-resolution imaging, the work provides convincing experimental data to support a model of "nucleoplasmic coarsening" between recombination intermediates and nucleoplasm for the control of crossover distribution in the context of a meiotic chromosome structure. The work will be of interest to researchers who work on meiosis as well as the regulation of chromosomal biology in general.

---

## [Decision Letter]

**Decision letter after peer review:**

Thank you for submitting your article "The synaptonemal complex controls cis- versus trans-interference in coarsening-based meiotic crossover patterning" for consideration by *eLife*. Your article has been reviewed by 3 peer reviewers, including Akira Shinohara as the Reviewing Editor and Reviewer #1, and the evaluation has been overseen by Jessica Tyler as the Senior Editor.

Essential revisions:

You propose a new model of the distribution of complexes involved in meiotic crossover formation in a synapsis-defective mutant and validated it by using published results and a new data set, to make your scientific claims more acceptable. The new model of nucleoplasmic coarsening is interesting and critical in the field of meiotic recombination. However, we strongly feel the additional analysis of your model using different genetic backgrounds would be required for further validation and application of your model as described below.

Moreover, given a recent preprint by Raphael Mercier (https://doi.org/10.1101/2022.05.11.491364), which shows similar results to your current and past results, it is essential to comment on the paper.

Please address the comments from the three reviewers, especially the four major points outlined below.

We would like to have your revision to be returned within two months. If the revision takes a lot longer, you may want to resubmit it as a new manuscript, and we will make sure that it goes to the same editors and reviewers.

1. To strengthen the idea of the "nucleoplasmic coarsening" model in the paper, in addition to the analysis of the zyp1a/1b mutant, it is important to check some genetic perturbation of key parameters in the model by changing such as a HEI10 amounts/concentration and/or the number of the binding sites (DSB or future CO sites). Given your group previously used HEI10 over-expressor (Morgan et al. Nature Communications, 2021), it is not difficult to see the effect of HEI10 over-expression in zyp1a/1b mutants on the model. Moreover, it is very critical to test your model using mutants that reduce the number of recombination intermediates such as asy1/+ and/or asy3/+ heterozygous mutants showing the altered distribution of COs and weakened interference (doi.org/10.1073/pnas.1921055117).

2. It would be nice if you integrate the "nucleoplasmic coarsening" model into the dynamics of recombination protein complexes in "wild-type" and "HEI10-overespressor" pachytene nuclei together with coarsening along chromosomes (Morgan et al. Nature Communications, 2021) to check the combination of the two types of the coarsening to explain wild-type (and HEI10-overespressor) CO distribution in a more robust way.

3. The term "trans-interference" is too strong and may be a bit misleading. The authors should be more careful to use trans-interference in the text including in the title of the paper. It would be better to use the other word with a detailed explanation.

4. It is essential to mention and compare the results described in the preprint by Mercier with the authors' results-how different and how similar the two papers are. This would provide much help to researchers in the related areas and readers of *eLife*.

*Reviewer #1 (Recommendations for the authors):*

For the publication, the authors need more experiments by changing the Hei10 dose, as shown in the original paper (Morgan, 2021), to validate the model described in the paper, and, if possible, need to integrate this new model (nucleoplasmic coarsening model) with previous one (cis-interference) to check whether combined regulatory mechanisms could explain CO distribution in wild-type plant (or not) in a more robust way. Moreover, the authors can apply combined simulation of cis and trans coarsening models in "wild-type" meiosis such as the zygotene stage (and early pachytene stage) in addition to the late pachytene stage where SC formation is limited to some chromosomal regions to explain the distribution of Hei10 in the early stage in wild-type meiosis with the number (more), intensity, and distribution of the foci.

The proposed model in this paper could explain the controlled localization of various proteins involved in meiotic recombination. It is attractive to check if the model could explain the localization of proteins involved in the recombination such as Msh4-5 complexes in the early pachytene stage.

In addition, it is now important to indicate that there is "little" cytoplasmic/nucleoplasmic Hei10 in wild-type pachytene nuclei (all Hei10 molecules in SC conduits/channels) experimentally.

The authors need to explain "trans-interference", which is a confusing word, more in detail such as "which interferes with which" (lines 141-157). More importantly, to avoid confusion, "trans-interference" should "be renamed" such as since it does not stand for what the authors analyzed here (sub-Poisson distribution of COs). This could be tested by increasing amounts of Hei10 in nucleoplasm as pointed out above. In addition, the authors discuss their observation of per-nucleus crossover covariation, which shows the broader distribution of COs in various organisms (Wang et al. Cell, 2017 & 2019)

Need staining analysis of Hei10 foci in early pachytene stages in the mutant as well as the late stage. In the earlier stage, the authors would see less bright and reduced numbers of foci.

*Reviewer #2 (Recommendations for the authors):*

Some specific points:

Line 1 (and throughout): It is not obvious that the meiosis field needs yet another term – "trans-interference" – to join the rather overcrowded field of terms describing statistical and mechanistic phenomena related to crossovers. Especially not one that is a compound of two terms that tend to be confusing in themselves. In addition, it is not clear, based on the current data, that what the authors name "trans-interference" indeed reflects a relevant biological entity and not a truly random distribution (the null hypothesis here).

Lines 19-20: what is the difference between 'quantitatively reproducing' and 'predicting' as pertaining to the work here? If there is no difference, one should be removed.

Line 143: What is the statistical test used to claim the "significance" referred to here? Crucially, statistical tests are missing throughout. Their absence is particularly notable here since this piece of data is crucial to the main conclusion of the manuscript.

Line 159: Figure 2D does not show what the authors claim – it merely shows many examples of coarsening, some stabilizing and some not by the end of the simulation. The manuscript would actually benefit from a more thorough analysis of this point since duration seems like a missed opportunity to test the model. What is the distribution of pachytene duration in plants? And how sensitive is the model to the distribution of this parameter?

Lines 197-217: The discussion of 'telomere loading' of recombination intermediates confuses underlying biological mechanisms and modeling strategy/approach. (And it is also confusing in general). This discussion needs to clearly indicate what is the biological reasoning behind the parameters being used. In its current form, it seems like the authors were simply fitting the model to the observed data. If that is the case, that should be clearly stated, and the statistical consequences of this addressed. A similar issue arises earlier, in lines 106-116, where it is not clear what was done to "fit the model" (line 106) and what were the findings that the "coarsening model was capable of recapitulating" (line 116).

Line 192 (and below): The term 'cytological recombination maps' (and the discussion of the genetic recombination maps from Capilla-Perez 2021) is confusing and misleading. The authors' quantitative cytological analysis is indeed novel and useful for their purposes, but it is not a 'recombination map'; it's a description of HEI10 foci. The two seem to be well correlated, but that does not mean they could be trivially equated. (A minor point, but in line 167, it should be noted that cytological analysis has not been done specifically *in zyp1 mutants*.)

Two final points:

First, as I'm sure the authors are well aware, a related manuscript from Raphael Mercier's group was placed in bioRxiv as this manuscript was under review (https://doi.org/10.1101/2022.05.11.491364). Please make sure to reference this preprint in the final form of your work.

Second, the clarity and readability of the manuscript will be greatly increased by limiting the number of abbreviations being used. Many of them are particularly uncommon or unique to the authors' own work (CP, RI).

*Reviewer #3 (Recommendations for the authors):*

The authors often refer to their previous model in which HEI10 coarsening is mediated via SC. The new model addresses the absence of SC, therefore HEI10 coarsening occurs via nucleoplasm. I think that it should be better emphasized in the work that the new model developed works only in a situation where SC is not present, while in the case of wild type and mutants where SC formation is not disturbed, the original model is applicable. I admit that it is well emphasized in the abstract, but not so clear later in the main body. This is particularly misleading in paragraphs that refer to Figure 4 (starting from line 191), where the authors present experimental data for both WT and zyp1, while showing the simulation for the mutant only. With a cursory reading, it is easy to lose this information somewhere and to think that the nucleoplasmic coarsening model can also be applied to WT. So why not show the simulation for WT using the original model in Figure 4 at the same time? I think it would improve the reception of work.

The paragraph on line 197 is difficult to follow and should be improved:

The sentence starting at Line 204: the authors should provide some figure explaining this effect of extra loading as it is not clear to me.

The sentence starting at line 210: when I tried to compare the original and nucleoplasmic coarsening models (Figure 2D in Nat commun paper and 4A in this ms), I couldn't see that end-loading is less pronounced in the new model than in the original one. Could you illustrate this more clearly and also show simulations from both models side-by-side?

The sentence starting at line 214: this is something I completely don't understand, as at the beginning of this paragraph you mentioned that in the zyp1 mutant HEI10 foci tend to be shifted toward the chromosome ends (which is also clearly visible in Figure 4A).

The way of using references in the manuscript is sometimes weird. E.g., I don't get why Capilla-Perez et al. 2021 is cited in line 110 and not just in 111 (this is the same sentence). In general, I would suggest including the references at the end of sentences and not in the middle of a sentence.

The methods are presented in a very clear and exhaustive way, I really appreciate this!

[Editors' note: further revisions were suggested prior to acceptance, as described below.]

Thank you for resubmitting your work entitled "Coarsening dynamics can explain meiotic crossover patterning in both the presence and absence of the synaptonemal complex" for further consideration by *eLife*. Your revised article has been evaluated by Detlef Weigel (Senior Editor) and a Reviewing Editor.

The manuscript has been improved but there are some remaining issues that need to be addressed, as outlined below:

In the revised version, the authors properly addressed our points by adding the results in the pch2 mutant and re-analyzing the published data (Durand 2022) and the manuscript has been improved. In summary, this paper provides a new model of the patterning of crossovers on meiotic chromosomes but will be accepted after some revision.

The study on HEI10 patterning in the pch2-1 mutant supports the SC-mediated and nucleocytoplasmic coarsening models. However, it is not clear to me why 40% of SC segments in the mutant show no HEI10 focus (line 359, Figure 5C) if the coarsening-mediated CO assurance functions on each segment. Does this mean a short SC segment does not have enough HEI10 molecules per segment to form a bright focus on RI? If so, the authors should show the classification of segment focus numbers based on the segment length category (e.g. short, middle, and long segments) as shown in Figure 5C. The analysis could give the minimum segment length for the CO assurance (only in the case that PCH2 does not play a direct role in CO formation). It is critical to explain this defect in the pch2 mutant based on the simulation parameters in the main text.

---

## [Author Response]

Essential revisions:You propose a new model of the distribution of complexes involved in meiotic crossover formation in a synapsis-defective mutant and validated it by using published results and a new data set, to make your scientific claims more acceptable. The new model of nucleoplasmic coarsening is interesting and critical in the field of meiotic recombination. However, we strongly feel the additional analysis of your model using different genetic backgrounds would be required for further validation and application of your model as described below.Moreover, given a recent preprint by Raphael Mercier (https://doi.org/10.1101/2022.05.11.491364), which shows similar results to your current and past results, it is essential to comment on the paper.Please address the comments from the three reviewers, especially the four major points outlined below.We would like to have your revision to be returned within two months. If the revision takes a lot longer, you may want to resubmit it as a new manuscript, and we will make sure that it goes to the same editors and reviewers.1. To strengthen the idea of the "nucleoplasmic coarsening" model in the paper, in addition to the analysis of the zyp1a/1b mutant, it is important to check some genetic perturbation of key parameters in the model by changing such as a HEI10 amounts/concentration and/or the number of the binding sites (DSB or future CO sites). Given your group previously used HEI10 over-expressor (Morgan et al. Nature Communications, 2021), it is not difficult to see the effect of HEI10 over-expression in zyp1a/1b mutants on the model. Moreover, it is very critical to test your model using mutants that reduce the number of recombination intermediates such as asy1/+ and/or asy3/+ heterozygous mutants showing the altered distribution of COs and weakened interference (doi.org/10.1073/pnas.1921055117).

To strengthen the empirical support for our mathematical modelling we have now added a substantial volume of additional experimental data. Specifically, we have added new experiments and accompanying modelling to investigate the patterning of late-HEI10 foci in *pch2* mutants, which exhibit incomplete synapsis and weakened interference. We have also demonstrated that our nucleoplasmic coarsening model can explain the recently published results showing that zyp1 mutation combined with HEI10 overexpression results in a massive increase in class I COs in *Arabidopsis* (Durand et al., 2022).

Whilst the reviewers initially suggested performing experiments in asy1 or asy3 heterozygotes, after additional consultation with the editor we were informed that we “may add the analysis of two mutants in your model (you may try mutants other than the asy3 heterozygotes).” We felt that a *pch2* mutant (alongside a *zyp1* mutant + HEI10 overexpressor, see below) offered the best opportunity to further test our model for the following reasons:

*pch2* mutants exhibit partial synapsis. This phenotype fits well within the narrative of the paper, where we have also examined the effect of coarsening in lines with full or completely absent synapsis. Partial synapsis effectively reduces the number of recombination intermediates that can mature into COs, as only the minority of recombination intermediates that are present at regions of synapsis can form COs.CO interference has already been shown, genetically, to be weaker in *pch2* mutants, both in *Arabidopsis* and other organisms (Lambing et al., 2015).DSB frequency does not appear to be affected in *Arabidopsis pch2* mutants (Lambing et al., 2015). Therefore, it is likely that the effects on CO patterning are a direct result of altered synapsis and coarsening, rather than due to the altered patterning of DSB precursors.

We believe the addition of this extra data, and the ability of the coarsening model to explain yet another non-trivial meiotic phenotype, substantially strengthens and supports our conclusions. We thank the reviewers for suggesting the inclusion of such experiments. In light of these additions, we have also made some very minor adjustments to our original nucleoplasmic coarsening model to ensure continuity of model parameters used in the different simulation outputs now presented within the paper. Additionally, it is important to note that the ability of the coarsening model to explain the effect of reduced DSB number on CO frequency was already tested within our previous publication (fitting model outputs to *spo11* hypomorph data from (Xue et al., 2018)).

2. It would be nice if you integrate the "nucleoplasmic coarsening" model into the dynamics of recombination protein complexes in "wild-type" and "HEI10-overespressor" pachytene nuclei together with coarsening along chromosomes (Morgan et al. Nature Communications, 2021) to check the combination of the two types of the coarsening to explain wild-type (and HEI10-overespressor) CO distribution in a more robust way.

As suggested, we have now added a section to the manuscript demonstrating that a model combining aspects of both synaptonemal complex (SC) and nucleoplasm-mediated coarsening can explain CO patterning in *Arabidopsis* wild-type and HEI10 overexpressor lines.

3. The term "trans-interference" is too strong and may be a bit misleading. The authors should be more careful to use trans-interference in the text including in the title of the paper. It would be better to use the other word with a detailed explanation.

After taking on board feedback from the reviewers, we have now removed the terms trans- and cisinterference from the manuscript. We have re-written the title and edited the text to shift the main focus of the paper towards the more general conclusions we have made, regarding the interplay between synapsis and coarsening.

4. It is essential to mention and compare the results described in the preprint by Mercier with the authors' results-how different and how similar the two papers are. This would provide much help to researchers in the related areas and readers of eLife.

We have now added additional modelling to the paper demonstrating that our nucleoplasmic coarsening model is fully capable of explaining the massive increase in class I COs observed in the (now fully published) paper by Durand et al. We have also included additional text in the introduction and Discussion sections to further discuss this paper.

Reviewer #1 (Recommendations for the authors):For the publication, the authors need more experiments by changing the Hei10 dose, as shown in the original paper (Morgan, 2021), to validate the model described in the paper, and, if possible, need to integrate this new model (nucleoplasmic coarsening model) with previous one (cis-interference) to check whether combined regulatory mechanisms could explain CO distribution in wild-type plant (or not) in a more robust way. Moreover, the authors can apply combined simulation of cis and trans coarsening models in "wild-type" meiosis such as the zygotene stage (and early pachytene stage) in addition to the late pachytene stage where SC formation is limited to some chromosomal regions to explain the distribution of Hei10 in the early stage in wild-type meiosis with the number (more), intensity, and distribution of the foci.

We have added extra modelling to the paper showing that the nucleoplasmic coarsening model can explain the massive increase in class I COs observed in *zyp1* mutants and HEI10 overexpressing *Arabidopsis* (Durand et al., 2022)*.* We have also added a new Results section that develops a combined coarsening model (incorporating SC and nucleoplasm-mediated coarsening) to explain CO patterning in wild-type and HEI10 overexpressing *Arabidopsis.* We have also investigated how this model operates in scenarios where SC formation is limited to only some chromosomal regions. However, instead of looking in zygotene nuclei, which would require detailed knowledge of the dynamics of synapsis initiation and elongation that is currently lacking, and which would dramatically increase model complexity, we have examined this scenario by including extra experiments and modelling in *pch2* mutants, which retain partial synapsis throughout the entirety of pachytene.

The proposed model in this paper could explain the controlled localization of various proteins involved in meiotic recombination. It is attractive to check if the model could explain the localization of proteins involved in the recombination such as Msh4-5 complexes in the early pachytene stage.

We do not have explicit data on MSH4-MSH5 localisation as, in our hands, we have not had much success with antibodies targeting these proteins, making it difficult to perform quantitative immunocytogenetics. However, it is important to note that the number of RIs in the initial conditions of our model is based on previously published counts of MSH4 and MSH5 foci (as well as early HEI10 foci) per cell (Higgins et al., 2008). Therefore, the expected frequency of these protein complexes is already incorporated within model parameters.

In addition, it is now important to indicate that there is "little" cytoplasmic/nucleoplasmic Hei10 in wild-type pachytene nuclei (all Hei10 molecules in SC conduits/channels) experimentally.

Unfortunately, to increase antibody penetration and specificity much of the cytoplasm and nucleoplasm is removed during the preparation of spread immunostained prophase I nuclei. This means that we cannot currently experimentally quantify the amount of SC bound vs nucleoplasmic HEI10. Live-cell imaging of fluorescently tagged HEI10 would be optimal for this experiment, but from personal communication with other labs that routinely perform live-cell imaging experiments, the tagging of HEI10 in *Arabidopsis* has proved to be experimentally challenging.

The authors need to explain "trans-interference", which is a confusing word, more in detail such as "which interferes with which" (lines 141-157). More importantly, to avoid confusion, "trans-interference" should "be renamed" such as since it does not stand for what the authors analyzed here (sub-Poisson distribution of COs). This could be tested by increasing amounts of Hei10 in nucleoplasm as pointed out above. In addition, the authors discuss their observation of per-nucleus crossover covariation, which shows the broader distribution of COs in various organisms (Wang et al. Cell, 2017 & 2019)

As described above, we have now removed the term trans-interference as it is clear this was a source of confusion. We have also added a small discussion of CO covariance and how recent evidence suggests this is absent in *Arabidopsis* (lines 398-402)*.*

Need staining analysis of Hei10 foci in early pachytene stages in the mutant as well as the late stage. In the earlier stage, the authors would see less bright and reduced numbers of foci.

We have now included a supplementary figure (Figure 3 —figure supplement 1) showing the staining of HEI10 in early pachytene cells from wild-type and *zyp1 Arabidopsis*. Indeed, we see less bright and *greater* (assuming this is what the reviewer meant to say?) numbers of foci.

Reviewer #2 (Recommendations for the authors):Some specific points:Line 1 (and throughout): It is not obvious that the meiosis field needs yet another term – "trans-interference" – to join the rather overcrowded field of terms describing statistical and mechanistic phenomena related to crossovers. Especially not one that is a compound of two terms that tend to be confusing in themselves. In addition, it is not clear, based on the current data, that what the authors name "trans-interference" indeed reflects a relevant biological entity and not a truly random distribution (the null hypothesis here).

As described above, we have now removed the terms trans- and cis- interference from the paper (we concede they were confusing!) and have toned down our prior emphasis on this aspect of the results.

Lines 19-20: what is the difference between 'quantitatively reproducing' and 'predicting' as pertaining to the work here? If there is no difference, one should be removed.

This is a good point and was not made sufficiently clear within the original manuscript. For model ‘predictions’ the model was not explicitly fitted to the data it explains. This is not the case for scenarios where the model ‘reproduced’ the experimental results, where the model *was* explicitly fitted to the available data. We have added additional text to clarify this point (lines 259-261).

Line 143: What is the statistical test used to claim the "significance" referred to here? Crucially, statistical tests are missing throughout. Their absence is particularly notable here since this piece of data is crucial to the main conclusion of the manuscript.

Details regarding the statistical tests used to show significant under-dispersion relative to a Poisson distribution have now been added (lines 147-152 and lines 172-185).

Line 159: Figure 2D does not show what the authors claim – it merely shows many examples of coarsening, some stabilizing and some not by the end of the simulation. The manuscript would actually benefit from a more thorough analysis of this point since duration seems like a missed opportunity to test the model. What is the distribution of pachytene duration in plants? And how sensitive is the model to the distribution of this parameter?

We have changed the wording of the sentence in the text that refers to this figure to clarify our meaning (lines 159-161). As the reviewer points out, the coarsening dynamics within our model are sensitive to the duration of pachytene, with longer durations resulting in fewer COs. The sensitivity of this parameter (and others) were tested within our previous publication by 10% perturbation, giving broadly comparable results (Morgan et al., 2021). Interestingly, the distribution of timings of meiosis (and, hence, likely pachytene) in plants does seem to vary considerably, with increased duration possibly correlating with increasing total SC length (Anderson et al., 1985; Bennett et al., 1977). We agree testing the impact of altered duration of meiosis on CO number and HEI10 dynamics offers an excellent opportunity to test the model, however we feel this lies outside the scope of this current paper, which is more focused on the interplay between synapsis and coarsening in *Arabidopsis*.

Lines 197-217: The discussion of 'telomere loading' of recombination intermediates confuses underlying biological mechanisms and modeling strategy/approach. (And it is also confusing in general). This discussion needs to clearly indicate what is the biological reasoning behind the parameters being used. In its current form, it seems like the authors were simply fitting the model to the observed data. If that is the case, that should be clearly stated, and the statistical consequences of this addressed. A similar issue arises earlier, in lines 106-116, where it is not clear what was done to "fit the model" (line 106) and what were the findings that the "coarsening model was capable of recapitulating" (line 116).

We have added additional text to clarify that the end-bias parameter was included to ‘improve the fit’ of the original coarsening model (lines 230-232). We think this makes it clear that this parameter was initially included to improve the fit of the model to the data, rather than for a predetermined biological reason (the inclusion of this parameter, and the effect of its absence, is discussed in greater detail in (Morgan et al., 2021)). We have also clarified that in the absence of this parameter we would not observe the distal bias of COs observed within our *zyp1* modelling (lines 232-233). However, we do believe the biological justifications for including these parameters are valid, which we discuss within the text. We have also added text clarifying how the model was fitted and specifically which data the model simulations recapitulate (lines 106-109 and lines 116-118).

Line 192 (and below): The term 'cytological recombination maps' (and the discussion of the genetic recombination maps from Capilla-Perez 2021) is confusing and misleading. The authors' quantitative cytological analysis is indeed novel and useful for their purposes, but it is not a 'recombination map'; it's a description of HEI10 foci. The two seem to be well correlated, but that does not mean they could be trivially equated. (A minor point, but in line 167, it should be noted that cytological analysis has not been done specifically *in zyp1 mutants*.)

This is a fair point. We have changed the terminology to ‘cytological late-HEI10 foci maps’ (line 219). We have also added the suggested text (line 190).

Two final points:First, as I'm sure the authors are well aware, a related manuscript from Raphael Mercier's group was placed in bioRxiv as this manuscript was under review (https://doi.org/10.1101/2022.05.11.491364). Please make sure to reference this preprint in the final form of your work.Second, the clarity and readability of the manuscript will be greatly increased by limiting the number of abbreviations being used. Many of them are particularly uncommon or unique to the authors' own work (CP, RI).

We have now referenced and discussed this (now fully published) work. We have also removed the abbreviation for CP from the manuscript, however we would prefer to keep RI (recombination intermediate) as this is found in some other publications and, importantly, maintains continuity with our previous coarsening paper (Morgan et al., 2021).

Reviewer #3 (Recommendations for the authors):The authors often refer to their previous model in which HEI10 coarsening is mediated via SC. The new model addresses the absence of SC, therefore HEI10 coarsening occurs via nucleoplasm. I think that it should be better emphasized in the work that the new model developed works only in a situation where SC is not present, while in the case of wild type and mutants where SC formation is not disturbed, the original model is applicable. I admit that it is well emphasized in the abstract, but not so clear later in the main body. This is particularly misleading in paragraphs that refer to Figure 4 (starting from line 191), where the authors present experimental data for both WT and zyp1, while showing the simulation for the mutant only. With a cursory reading, it is easy to lose this information somewhere and to think that the nucleoplasmic coarsening model can also be applied to WT. So why not show the simulation for WT using the original model in Figure 4 at the same time? I think it would improve the reception of work.

We believe that with the addition of the section discussing and exploring a combined version of the model, that incorporates SC and nucleoplasm-mediated coarsening, it will be much clearer that the full nucleoplasmic model can only be used for SC mutants, whilst a model incorporating only a small amount of nucleoplasmic coarsening can explain CO patterning in wild-type *Arabidopsis*. Additionally, we have now incorporated wild-type CO patterning simulations in Figure 4 —figure supplement 2, which readers will be able to compare with the data in Figure 4.

The paragraph on line 197 is difficult to follow and should be improved:The sentence starting at Line 204: the authors should provide some figure explaining this effect of extra loading as it is not clear to me.

We have added some additional text (lines 224-248) and an additional supplementary figure (Figure 1 —figure supplement 1) to clarify our meaning in this section.

The sentence starting at line 210: when I tried to compare the original and nucleoplasmic coarsening models (Figure 2D in Nat commun paper and 4A in this ms), I couldn't see that end-loading is less pronounced in the new model than in the original one. Could you illustrate this more clearly and also show simulations from both models side-by-side?

This has been added in the new supplementary figure, described above (Figure 1 —figure supplement 1).

The sentence starting at line 214: this is something I completely don't understand, as at the beginning of this paragraph you mentioned that in the zyp1 mutant HEI10 foci tend to be shifted toward the chromosome ends (which is also clearly visible in Figure 4A).

We have reworded this sentence (lines 239-242).

The way of using references in the manuscript is sometimes weird. E.g., I don't get why Capilla-Perez et al. 2021 is cited in line 110 and not just in 111 (this is the same sentence). In general, I would suggest including the references at the end of sentences and not in the middle of a sentence.

We have modified the referencing of Capilla-Perez et al., 2021 within the manuscript, as suggested.

The methods are presented in a very clear and exhaustive way, I really appreciate this!

Thank you!

References:

Anderson LK, Stack SM, Fox MH, Chuanshan Z. 1985. The relationship between genome size and synaptonemal complex length in higher plants. *Exp Cell Res* 156:367–378. doi:https://doi.org/10.1016/0014-4827(85)90544-0

Bennett MD, Lewis KR, Harberd DJ, Riley R, Bennett MD, Flavell RB. 1977. The time and duration of meiosis. *Philosophical Transactions of the Royal Society of London B, Biological Sciences* 277:201–226. doi:10.1098/rstb.1977.0012

Capilla-Pérez L, Durand S, Hurel A, Lian Q, Chambon A, Taochy C, Solier V, Grelon M, Mercier R. 2021. The synaptonemal complex imposes crossover interference and heterochiasmy in Arabidopsis. *Proceedings of the National Academy of Sciences* 118:e2023613118. doi:10.1073/pnas.2023613118

Durand S, Lian Q, Jing J, Ernst M, Grelon M, Zwicker D, Mercier R. 2022. Joint control of meiotic crossover patterning by the synaptonemal complex and HEI10 dosage. *Nat Commun* 13:5999. doi:10.1038/s41467-022-33472-w

Higgins JD, Vignard J, Mercier R, Pugh AG, Franklin FCH, Jones GH. 2008. AtMSH5 partners AtMSH4 in the class I meiotic crossover pathway in *Arabidopsis thaliana*, but is not required for synapsis. *The Plant Journal* 55:28–39. doi:10.1111/j.1365-313X.2008.03470.x

Lambing C, Osman K, Nuntasoontorn K, West A, Higgins JD, Copenhaver GP, Yang J, Armstrong SJ, Mechtler K, Roitinger E, Franklin FCH. 2015. Arabidopsis PCH2 Mediates Meiotic Chromosome Remodeling and Maturation of Crossovers. *PLoS Genet* 11:e1005372-.

Morgan C, Fozard JA, Hartley M, Henderson IR, Bomblies K, Howard M. 2021. Diffusion-mediated HEI10 coarsening can explain meiotic crossover positioning in Arabidopsis. *Nat Commun* 12:4674. doi:10.1038/s41467-021-24827-w

Xue M, Wang J, Jiang L, Wang M, Wolfe S, Pawlowski WP, Wang Y, He Y. 2018. The number of meiotic double-strand breaks influencecrossover distribution in arabidopsis[open]. *Plant Cell*. doi:10.1105/tpc.18.00531

[Editors' note: further revisions were suggested prior to acceptance, as described below.]

The manuscript has been improved but there are some remaining issues that need to be addressed, as outlined below:In the revised version, the authors properly addressed our points by adding the results in the pch2 mutant and re-analyzing the published data (Durand 2022) and the manuscript has been improved. In summary, this paper provides a new model of the patterning of crossovers on meiotic chromosomes but will be accepted after some revision.The study on HEI10 patterning in the pch2-1 mutant supports the SC-mediated and nucleocytoplasmic coarsening models. However, it is not clear to me why 40% of SC segments in the mutant show no HEI10 focus (line 359, Figure 5C) if the coarsening-mediated CO assurance functions on each segment. Does this mean a short SC segment does not have enough HEI10 molecules per segment to form a bright focus on RI? If so, the authors should show the classification of segment focus numbers based on the segment length category (e.g. short, middle, and long segments) as shown in Figure 5C. The analysis could give the minimum segment length for the CO assurance (only in the case that PCH2 does not play a direct role in CO formation). It is critical to explain this defect in the pch2 mutant based on the simulation parameters in the main text.

Thank you for raising this interesting point. We have now added additional text to this Results section to further discuss why zero CO segments occur within our *pch2* simulations (lines 362-367). We have also edited Figure 5 – S1 to make it clearer that some short SC segments lack any RIs in our *pch2* simulations. We note that the classification of segment focus numbers based on segment SC length is already shown in Figure 5F. However, we have now added additional text to discuss the important point raised by the reviewer regarding the minimum SC length required for CO assurance in *pch2* mutants (lines 330-332).